# ErbB signaling is a potential therapeutic target for vascular lesions with fibrous component

Suvi Jauhiainen[1†], Henna Ilmonen[1†], Pia Vuola[2,3], Heta Rasinkangas[1], Heidi H Pulkkinen[1], Sara Keränen[1], Miika Kiema[1], Jade J Liikkanen[1], Nihay Laham-Karam[1], Svetlana Laidinen[1], Mustafa Beter[1], Einari Aavik[1], Kimmo Lappalainen[3,4], Jouko Lohi[3,5], Johanna Aronniemi[3,4], Tiit Örd[1], Minna U Kaikkonen[1], Päivi Salminen[3,6], Erkki Tukiainen[2], Seppo Ylä-Herttuala[1,7,8], Johanna P Laakkonen[1]*

[1]A.I. Virtanen Institute for Molecular Sciences, University of Eastern Finland, Kuopio, Finland; [2]Department of Plastic Surgery, Helsinki University Hospital and University of Helsinki, Helsinki, Finland; [3]VASCERN VASCA European Reference Centre, Helsinki University Hospital, Helsinki, Finland; [4]Department of Radiology, HUS Diagnostic Center and Helsinki University Hospital and University of Helsinki, Helsinki, Finland; [5]Department of Pathology, HUSLAB, Helsinki University Hospital and University of Helsinki, Helsinki, Finland; [6]Department of Pediatric Surgery, New Children's Hospital, Helsinki University Hospital and University of Helsinki, Helsinki, Finland; [7]Science Service Center, Kuopio University Hospital, Kuopio, Finland; [8]Gene Therapy Unit, Kuopio University Hospital, Kuopio, Finland

**\*For correspondence:**
johanna.p.laakkonen@uef.fi

[†]These authors contributed equally to this work

**Competing interest:** The authors declare that no competing interests exist.

## Abstract

**Background:** Sporadic venous malformation (VM) and angiomatosis of soft tissue (AST) are benign, congenital vascular anomalies affecting venous vasculature. Depending on the size and location of the lesion, symptoms vary from motility disturbances to pain and disfigurement. Due to the high recurrence of the lesions, more effective therapies are needed.

**Methods:** As targeting stromal cells has been an emerging concept in anti-angiogenic therapies, here, by using VM/AST patient samples, RNA-sequencing, cell culture techniques, and a xenograft mouse model, we investigated the crosstalk of endothelial cells (EC) and fibroblasts and its effect on vascular lesion growth.

**Results:** We report, for the first time, the expression and secretion of transforming growth factor A (TGFA) in ECs or intervascular stromal cells in AST and VM lesions. TGFA induced secretion of vascular endothelial growth factor (VEGF-A) in paracrine fashion, and regulated EC proliferation. Oncogenic *PIK3CA* variant in p.H1047R, a common somatic mutation found in these lesions, increased TGFA expression, enrichment of hallmark hypoxia, and in a mouse xenograft model, lesion size, and vascularization. Treatment with afatinib, a pan-ErbB tyrosine-kinase inhibitor, decreased vascularization and lesion size in a mouse xenograft model with ECs expressing oncogenic *PIK3CA* p.H1047R variant and fibroblasts.

**Conclusions:** Based on the data, we suggest that targeting of both intervascular stromal cells and ECs is a potential treatment strategy for vascular lesions having a fibrous component.

**Funding:** Academy of Finland, Ella and Georg Ehnrooth foundation, the ERC grants, Sigrid Jusélius Foundation, Finnish Foundation for Cardiovascular Research, Jane and Aatos Erkko Foundation, GeneCellNano Flagship program, and Department of Musculoskeletal and Plastic Surgery, Helsinki University Hospital.

## Editor's evaluation

The authors have explored potential crosstalk between endothelial cells and fibroblasts in the context of sporadic vascular malformation (venous malformation and angiomatoses of soft tissue). With a high level of evidence, they found that mutated endothelial cells secrete TGFa that will activate surrounding fibroblasts, leading in turn to VEGFA secretion that will stimulate endothelial cells sprouting and vascular malformation development. This is an important advance in the field.

## Introduction

Sporadic VM and AST form a heterogeneous group of vascular anomalies affecting venous vasculature (*Merrow et al., 2016*; *Mulliken et al., 2013*). Lesions form due to a local defect in vascular development during embryogenesis and expand with time, manifesting clinically usually in late childhood or early adulthood (*Hassanein et al., 2012*). VM can locate in any tissue or internal organ and be either superficial or permeate multiple tissue planes (*Dompmartin et al., 2010*), whereas AST is typically found in extremities or trunks being subcutaneous or intramuscular. In both, symptoms vary from limited aesthetic harm to motility disturbances, muscle weakness, pain, disfigurement, and life-threatening bleeding, depending on the size and location of the lesion.

Overlapping magnetic resonance imaging findings, but distinctive histological features, are usually found between VM and AST (*Aronniemi et al., 2017*). In both AST and VM, venous structures form enlarged, irregular vascular channels (*Aronniemi et al., 2017*; *Brouillard and Vikkula, 2003*). Fibrous connective tissue with fibroblasts is detected around the various-sized vessels in AST (*Aronniemi et al., 2017*), while sclerotherapy can cause secondary fibrosis in VMs (*Aronniemi et al., 2016*). Whereas VMs consists solely of venous structures, AST also has artery-like vessels, lymphatic vessels, small capillaries and mesenchymal tissue components, especially muscle-infiltrating fat (*Aronniemi et al., 2017*; *Rao and Weiss, 1992*) In ISSVA classification (i.e. International Society for the Study of Vascular Anomalies), AST is classified under provisionally unclassified vascular anomalies (issva.org/classification, y.2018). In sporadic VMs, somatic mutations in tyrosine-protein kinase receptor *TEK* are found in approximately half of the patients, while somatic *PIK3CA* mutations are found in 20% of the VMs lacking *TEK* alterations (*Limaye et al., 2015*; *Soblet et al., 2013*). Somatic *PIK3CA* mutations have also been associated with AST (*Boccara et al., 2020*). Causative mutations in these genes lead to chronic activation of AKT and dysregulation of EC migration, expression of angiogenic factors as well as alterations in composition and processing of the extracellular matrix (*Limaye et al., 2015*; *Nätynki et al., 2015*; *Uebelhoer et al., 2013*).

VM or AST do not regress spontaneously. If conservative treatment is ineffective, symptomatic lesions are treated with percutaneous sclerotherapy, percutaneous cryotherapy, endovascular laser treatment, or surgical resection (*Rosenblatt, 2007*; *Steiner et al., 2013*; *Hage et al., 2018*; *Cornelis et al., 2017*; *Patel et al., 2017*). At present, sirolimus targeting the PI3K/AKT/mTOR pathway is tested in clinical trials for the treatment of VM (*Adams et al., 2016*; *Boscolo et al., 2015*) (ClinicalTrials.gov, study nro: NCT02638389). So far, most of the previous studies have been focused on the role of ECs in VM or AST pathogenesis. As targeting of stromal cells (SCs) is an emerging concept for the development of anti-angiogenic therapies (*Bussard et al., 2016*; *Donnem et al., 2008*; *Criscitiello et al., 2014*), we studied here crosstalk of ECs and intervascular stromal cells in VM and AST and assessed the role of fibroblasts in PI3K-driven lesion growth in mice.

## Materials and methods

### Patient cohort

The multidisciplinary vascular anomaly team of Helsinki University Hospital (HUS) evaluated the patients clinically and radiologically and selected the treatment line. Patient samples (*Tables 1 and 2*) were collected in elective surgery in the Department of Plastic Surgery, Helsinki, HUS, Helsinki, Finland. A decision for surgical treatment was based on clinical practices. Patient sample collection was approved by the Ethical Committee of the HUS (Decision No 127/13/03/02/2010 and No 1394/2020). Informed consent was obtained from all patients included in the study. Samples were studied by a pathologist (JL) who specialized in vascular anomalies and classified according to ISSVA guidelines

**Table 1.** Demographics of patients with AST patients.

| Patient | Gender | Age | Pathological diagnosis | # of lesions | Tissue* | Location | Somatic mutation (Fractional abundance §) |
|---|---|---|---|---|---|---|---|
| 1 [†,‡] | F | 11 | AST | 1 | im | calf | ND |
| 2 [†,‡] | M | 34 | AST | 1 | im | shoulder | PI3KCA p.E542K (WTL: 10.25; EC: 50.65; SC: none) |
| 3 [†,†] | M | 16 | AST | 1 | im | calf | PI3KCA p.H1047R (WTL: 18.80; EC: 44.00; SC: none) |
| 4 [†,‡] | F | 16 | AST | 1 | sc | shin | PI3KCA p.H1047L (WTL: 8.30; EC: 48.95; SC: none) |
| 5 [†] | F | 17 | AST | 1 | im | back | ND |
| 6 | M | 34 | AST | 1 | im | thigh | ND |
| 7 | M | 17 | AST | 1 | im | thigh | PI3KCA p.E542K (WTL: 7.32) |
| 8 | F | 31 | AST | 1 | im | thigh | PI3KCA p.H1047R (WTL: 5.07) |
| 9 | F | 22 | AST | 1 | im | thigh | PI3KCA p.H1047R (WTL: 13.10) |
| 10 | F | 13 | AST | 1 | im, sc | foot | PI3KCA p.E545K (WTL: 11.65) |
| 11 | M | 19 | AST | 1 | im | thigh | PI3KCA p.H1047R (WTL: 8.89) |
| 12 | M | 13 | AST | 1 | im, sc | ankle | PI3KCA p.H1047R (WTL: 12.60) |
| 13 | F | 23 | AST | 1 | im | calf | PIK3CA p.Y644H [¶] |
| 14 | F | 41 | AST | 1 | sc | shin | ND |
| 15 | F | 25 | AST | 1 | im | foot | PI3KCA p.E545K (WTL: 8.93) |
| 16 | F | 16 | AST | 1 | im, sc | ankle | PI3KCA p.H1047R (WTL: 5.98) |
| 17 | F | 46 | AST | 1 | im, sc | back | PI3KCA p.E542K (WTL: 9.83) |
| 18 | F | 18 | AST | 1 | im, sc | ankle | PI3KCA p.E545K (WTL: 11.35) |
| 19 | F | 13 | AST | 1 | im | calf | ND |
| 20 | F | 24 | AST | 1 | im, sm | thigh | PI3KCA p.E542K (WTL: 5.64) |

*Tissue: im, intramuscular; sc, subcutaneous; sm, synovial membrane.
[†]ECs and SCs isolated for cell experiments.
[‡]used in RNA-seq experiment.
[§]Fractional abundance of the mutation in WTL, whole tissue lysate; EC, endothelial cells; SC, intervascular stromal cells; ND, not detected.
[¶]mutation detected by whole-exome sequencing.

by using hematoxylin-eosin staining and immunohistochemistry (Glut-1, CD31, CD34, and D2-40). For DNA, RNA and protein work, tissue samples were taken immediately after resection from the middle of the lesion, snap-frozen in liquid nitrogen, and stored at –70 °C. Optionally, tissue samples were fixed with 4% paraformaldehyde for immunohistochemical staining or collected to Dulbecco's Modified Eagles' Medium (DMEM; Sigma-Aldrich, St. Louis, MOK, USA) supplemented with 20% Fetal Bovine Serum (FBS), 20 mM HEPES and antibiotics for cell isolation. Control tissue samples were normal vascular specimens (from the mammary artery, n=4; or saphenous vein n=2) from atherosclerotic patients undergoing bypass surgery. After removal, tissue material not needed for a bypass graft was snap-frozen in liquid nitrogen and used for research purposes with approval from the Research Ethics Committee of the Northern Savo Hospital District (Decision No 139/2015).

## Cell culture

Resected patient tissue samples were treated with collagenase type II (Worthington, Lakewood, NJ, USA) for an hour at 37 °C under agitation. Selection of ECs was performed with CD31 MicroBead Kit and a magnetic column (Miltenyi Biotec, Bergisch Gladbach, Germany) as previously described (*Partanen et al., 2013*). Patient-derived ECs were cultured on fibronectin/gelatin-coated cell culture flasks in Endothelial Cell Growth medium (EGM; Cambrex Biosciences, East Rutherford, NJ, USA) supplemented with 20% FBS. Patient-derived intervascular SCs from flow-through fraction were

**Table 2.** Demographics of patients with VM.

| Patient | Gender | Age | Pathological diagnosis | # of lesions | Tissue* | Location | Somatic mutation (Fractional abundance in whole tissue lysate) § |
|---|---|---|---|---|---|---|---|
| 21 | M | 34 | VM | 1 | im | thigh | *TEK* p.Y1108X ¶ |
| 22 ,†, ‡ | F | 77 | VM | 6 | sc | neck, fossa cubitalis, chest, hip, big toe | *TEK* p.L914F (5.63) |
| 23 | M | 40 | VM | 2 | im | chest, back | ND |
| 24 | F | 69 | VM | 3 | im, sc | forearm **, hand | *TEK* p.L914F (10.01) |
| 25 | M | 24 | VM | 2 | sc | ankle, sole | *PI3KCA* p.H1047L (3.83) |
| 26 | M | 14 | VM | 1 | sc | lip | *TEK* p.L914F (16.31) |
| 27 | F | 39 | VM | 1 | im | chest | ND |
| 28 | M | 28 | VM | 1 | sc | clavicle | *TEK* p.L914F (5.95) |
| 29 | M | 46 | VM | 1 | sc | shin | ND |
| 30 | M | 31 | VM | 1 | sc | knee | *TEK* p.L914F (10.60) |
| 31 | F | 21 | VM | 1 | im, sc, sm | knee | ND |
| 32 | M | 16 | VM | 1 | im, sc, sm | thigh, knee | ND |
| 33 | M | 9 | VM | 1 | sc, sm | knee | ND |
| 34 | F | 35 | VM | 1 | im, sc | blade | *TEK* p.L914F (12.10) |
| 35 | F | 16 | VM | 1 | sc | ankle | *PI3KCA* p.E545K (4.34) |
| 36 | M | 21 | VM/AST | 1 | im | upper arm | *PI3KCA* p.H1047R (3.91) |
| 37 | F | 41 | VM/AST | 1 | sc | ankle | *KRAS* p.Q61R ¶ |
| 38 | F | 25 | VM/AST | 1 | sc | calf, leg | *PI3KCA* p.H1047L (5.23) |

*Tissue: im, intramuscular; sc, subcutaneous; sm, synovial membrane.

† ECs and SCs isolated for cell experiments.

‡used in RNA-seq experiment.

§ND, not detected.

¶mutation detected by whole-exome sequencing.

**The patient had multiple lesions but only a lesion located in the forearm was operated.

maintained in DMEM supplemented with 10% FBS and antibiotics. HUVECs were isolated from human umbilical cords with approval from the Research Ethics Committee of the Northern Savo Hospital District, Kuopio, Finland (Decision No 341/2015) as previously described (*Jaffe et al., 1973*) and maintained in EGM. Human saphenous vein endothelial cells (HsaVEC) and control human pulmonary fibroblasts (HPF-c) were obtained from PromoCell (three donors/each, Heidelberg, Germany) and maintained according to manufacturer's instructions in EGM supplemented with 20% FBS or in DMEM supplemented with 10% FBS and antibiotics, respectively. Selection of control ECs was performed by CD31 MicroBead Kit. Patient SCs were characterized prior to experiments by western blot showing to be negative for EC marker CD31, and positive for fibroblast and smooth muscle cell marker vimentin. Additionally, alpha-smooth muscle cell actin, a marker of both myofibroblasts and smooth muscle cells, was detected in two cell lines (*Figure 2—figure supplement 2*).

## Lentivirus vectors

pHAGE-PIK3CA encoding *PIK3CA* wild-type (*PIK3CA*wt; Addgene plasmid #116771; http://n2t.net/addgene:116771; RRID:Addgene_116771) and pHAGE-PIK3CA-H1047R encoding *PIK3CA* with oncogenic point mutation on p.H1047R (*PIK3CA*H1047R; Addgene plasmid #116500; http://n2t.net/addgene:116500; RRID:Addgene_116500) were received as gifts from Gordon Mills & Kenneth Scott (*Ng et al., 2018*). Third-generation lentiviruses were produced in National Virus Vector Laboratories (NVVL, UEF, Kuopio, Finland). For experiments, HUVECs were seeded onto six-well plates at a density of

125.000 cells/well and allowed to adhere for 4 hr. Cells were transduced in fresh media with lentivirus vectors expressing $PIK3CA^{wt}$ or $PIK3CA^{H1047R}$ with the multiplicity of infection (MOI) of 7.5–10. After culturing the cells for 16 hr, cells were washed with PBS (Thermo Fisher Scientific, Waltham, MA, USA) and a fresh growth medium was added. After 72 hr cells were passaged onto new six-well plates and culturing continued for an additional 72 hr after which cells were harvested in Buffer RLT (Qiagen; for RNA-sequencing, RT-qPCR) or used for cell culture experiments.

## RNA-sequencing and gene ontology analysis

Total RNA of ECs was isolated using RNeasy Mini Kit according to the manufacturer's instructions (Qiagen, Hilden, Germany).

For patient-derived cell cultures, preparation of RNA-Seq libraries as well as data analysis for differentially expressed genes was performed as previously described (*Laakkonen et al., 2017*). Briefly, Poly(A)-RNA was enriched with MicroPoly(A) Purist Kit, fragmented using RNA Fragmentation Reagents (Thermo Fisher Scientific), and purified by running through P-30 column (Bio-Rad Laboratories, Hercules, CA, USA). The 3′ ends of the fragmented RNA was dephosphorylated with T4 polynucleotide kinase (PNK, New England Biolabs, Ipswich, MA, USA) followed by heat-inactivation. Dephosphorylation reactions were purified using anti-BrdU beads (Santacruz Biotech, Heidelberg, Germany) and precipitated overnight. Poly(A)-tailing and cDNA synthesis were performed the next day. After cDNA synthesis, Exonuclease I (New England Biolabs) was used to catalyze the removal of excess oligos. The DNA-RNA hybrid was purified using ChIP DNA Clean & Concentrator Kit (Zymo Research Corporation, Irvine, CA, USA), RNaseH treated, and circularized. The libraries were amplified for 11–14 cycles with the oNTI201-primer and a barcode-specific primer oNTI200-index. The final product was run on Novex 10% TBE gel, purified, and cleaned up as above. The libraries were sequenced on the Illumina Genome Analyzer 2 or HiSeq 2000 according to the manufacturer's instructions (GeneCore, EMBL, Heidelberg, Germany). RNA-seq was mapped using TopHat (v2.0.7). Poor quality reads were filtered out (minimum 97% of bp over quality cutoff 10) and the tag per base value was set to 3. Differentially expressed genes were identified using edgeR (*Robinson et al., 2010*).

For lentivirus experiments, RNA-Seq libraries were prepared from total RNA using the QuantSeq 3′ mRNA-Seq Library Prep Kit FWD for Illumina (Lexogen, Vienna, Austria) according to the manufacturer's instructions. The libraries were sequenced with a read length of 68 bp (single end) on an Illumina NextSeq 500 sequencer. The RNA-Seq reads were processed using the nf-core RNA-Seq pipeline (version 3.0) (*Ewels et al., 2020*) with the GRCh37 genome and the default quantification workflow (STAR aligner for read mapping and Salmon for gene quantification), followed by DESeq2 (version 1.22.2) (*Love et al., 2014*) differential expression analysis.

Each sequencing experiment was normalized to a total of $10^7$ uniquely mapped tags and visualized by preparing custom tracks for the UCSC Genome browser. Clustering results were generated by Cluster 3.0 (*de Hoon et al., 2004*) by normalizing and centering the gene expression tags to range from −1–1. The following thresholds were used: FDR <0.1 (Patient$^{CD31+}$EC vs HUVEC), p-value <0.05 (Patient $^{CD31+}$EC vs HsaVEC), FDR-adjusted p-value <0.1 ($PIK3CA^{H1047R}$ vs $PIK3CA^{wt}$ -transduced ECs and $PIK3CAmut+$ Patient $^{CD31+}$EC vs Ctrl ECs), RPKM >0.5 and log2 fold changes >1.0 and <−1.0. For gene ontology analysis, HOMER 4.3. or the EnrichR web server was used (*Chen et al., 2013*; *Heinz et al., 2010*; *Kuleshov et al., 2016*; *Xie et al., 2021*). Gene Set Enrichment Analysis (GSEA; https://www.biorxiv.org/content/10.1101/060012v3) with a custom gene set calling was used to compare the similarity of gene expression patterns between separate experiments. Motif enrichment was analyzed from the merged list of H3K4me2- and H3K27ac-defined enhancers that were located within 100 kb of the transcriptional start site (TSSs) of the differentially expressed genes. The 'findMotifsGenome.pl' command in the HOMER software was used with default settings, the peak size of 200 bp, and motif length of 8, 10, and 12 bases. A random set of genomic positions matched for GC% content was used as background. Enhancer elements enriched for H3K4me2 and H3K27ac marks in HUVECs (data from GSE29611) were generated using the 'findPeaks' command in the HOMER software (*Heinz et al., 2010*) with default settings for the 'style histone' option: identification of 500 bp regions, fourfold enrichment over input tag count 4, 0-fold enrichment over local tag count and 0.001 FDR significance. To select the coordinates of enhancers within 100 kb of the TSS 'mergePeaks' command with -cobound 1 and -d 100,000 were used.

**Table 3.** Selected cell signaling pathways regulated in patient-derived ECs.

| GO | Term | Genes* |
|---|---|---|
| GO:0038127 | ERBB signaling pathway | *PRKCE,TNRC6C,PDE1A,**AREG**, RPS27A,**TGFA**,KITLG,**ERBB2**, PTK2B,DGKD,PRKAR2B,**NRG1**, RPS6KA5,PRKACB* |
| GO:0043122 | regulation of I-kappaB kinase/ NF-kappaB signaling | *GREM1,LURAP1,BIRC3,TNFAIP3, S100A4,MALT1,LITAF,PRKCE, LPAR1,C18orf32,ZFAND6,RPS27A, PLK2,TLR4,F2RL1,CASP1,S100A13* |
| GO:0008277 | regulation of G-protein coupled receptor protein signaling pathway | *RGS4,RGS9,DYNLT1,RGS11,CXCL8,RGS10,RGS20,RGS5,RAMP2,RGS7, PLCB1,ADRBK2* |
| GO:0043410 | positive regulation of MAPK cascade | *PAK1,MAP4K2,SEMA4C,**TGFA**, GADD45G,KSR2,**ERBB2**,NENF, PLCB1,TPD52L1,GLIPR2,ICAM1, CD74,PRKCE,LPAR1,PDCD10,INSR, IGF1R,GADD45A,RPS27A,PTK2B, KITLG,HGF,F2RL1,TLR4,PIK3CG, ZEB2,**EPGN*** |
| GO:0046328 | regulation of JNK cascade | *PAK1,MAP4K2,GADD45A,IGF1R, GADD45G,TNXB,PTK2B,CBS,SFRP1,PLCB1,F2RL1,TLR4,TPD52L1,ZEB2* |
| GO:0007219 | Notch signaling pathway | *HEY2,NOTCH1,TNRC6C,RBX1, TMEM100,LFNG,RPS27A,E2F1, DTX3,NOTCH2,MESP1,DNER, FOXC1,SNAI2,HDAC9* |

*ErbB pathway receptors and ligands are marked in bold.

RNA-seq data have been submitted to NCBI Gene Expression Omnibus under accession numbers GSE130807 and GSE196311 (GEO reviewer access tokens; wbivkayaxhojdqp and mbehiikgvtmfryh, respectively). A summary of the NGS samples and gene lists is found in *Table 3* and *Supplementary file 1*.

## qRT-PCR

Total RNA was isolated from control/patient cells and tissue samples either with RNeasy Mini Kit (Qiagen, Hilden, Germany) or Tri Reagent according to the manufacturer's instructions (Sigma-Aldrich). cDNA synthesis and qRT-PCR were performed using target gene-specific Taqman assays (ThermoFisher Scientific, *Table 4*). Amplification of beta-2 microglobulin (*B2M*; for tissue samples) or glyceraldehyde-3-phosphate dehydrogenase (*GAPDH*; for ECs, HPF-c, and patient SCs) was used as an endogenous control to standardize the amount of RNA in each sample. Detection was performed with StepOnePlus Real-Time PCR System (Applied Biosystems, Foster City, CA, USA).

**Table 4.** Taqman assays used in qRT-PCR analysis.

| Gene | Description | Assay ID |
|---|---|---|
| *GAPDH* | Glyceraldehyde-3-phosphate dehydrogenase | 4352934E |
| *B2M* | Beta-2-microglobulin | Hs00187842_m1 |
| *TGFA* | Transforming growth factor A [assay 1] | Hs00608187_m1 |
| *TGFA* | Transforming growth factor A [assay 2] | Hs00177401_m1 |
| *TGFA* | Transforming growth factor A [assay 3] | HsaCEP0053322 |
| *ERBB1* | Protein tyrosine kinase ERBB1, epidermal growth factor receptor | Hs01076090_m1 |
| *AREG* | Amphiregulin | Hs00950669_m1 |
| *NRG1* | Neuregulin 1 | Hs00247620_m1 |
| *EPGN* | Epithelial mitogen, epigen | Hs02385424_m1 |
| *VEGF-A* | Vascular endothelial growth factor A | Hs00900055_m1 |
| *PIK3CA* | Phosphatidylinositol-4,5-Bisphosphate 3-Kinase Catalytic Subunit-α | HsaCEP0050716 |

## Recombinant proteins and inhibitors

Cells were seeded on six-well plates at the density of 200.000 cells/well. When cells reached 80% confluency they were washed with PBS and synchronized with basal media containing 0.5% FBS. After 16 hr 50 ng/ml of recombinant human (rh)TGFA (Sigma-Aldrich) and/or Afatinib (5 µM, MedChem Express, Monmouth Junction, NJ) was added to the wells. The corresponding concentration of DMSO was used as a control for Afatinib.

## ELISA

Expression levels of TGFA and VEGF-A in cell culture supernatants were measured using Human Quantikine ELISAs (R&D Systems, Minneapolis, MN, USA) according to the manufacturer's instructions. Due to the limited availability of patient-derived cells, mRNA expression and protein secretion analysis with patient samples were done on the same wells. Thus, protein concentration measured from cell culture medium was normalized to total RNA extracted from the same well at the same time point.

## Fibrin bead assay

Fibrin bead assay for HUVECs and HPF-c cells has been previously described (*Nakatsu et al., 2007*; *Pulkkinen et al., 2021*). Here, cytodex microcarrier beads were coated with HUVECs and embedded into a fibrin gel. HPF-c cells or, for the first time in this study, patient-derived intervascular stromal$^{CD31-,}$ $^{vimentin+}$ cells were layered on top of the gel with or without rhTGFA, rhVEGF-A (R&D Systems) or their combination (50 ng/ml each). Culturing was continued by changing a fresh EGM ±growth factors every other day. Afatinib (5 µM) or DMSO was added to the wells on day 3 and day 5. On day 7, the HPF-c layer on top of the fibrin gel was removed by trypsinization. ECs inside of the gel were fixed, permeabilized, and stained with phalloidin-A635 (F-actin, Thermo Fisher Scientific) and DAPI. Imaging was performed using LSM800 (Zeiss). 405/555 nm diode lasers were used together with the appropriate emission filters (10 x/0.3 PlanApo objective, 512 × 512 frame size). Image processing and quantitative analysis were performed from 3D-images by ImageJ (*Abràmoff et al., 2004*), in a blinded manner by two independent observers. Sprouts containing >1 nuclei were included in the analysis. Segmented vascular area was additionally detected.

## siRNA transfection, followed by imaging of cell growth with IncuCyte

HUVECs expressing *PIK3CA*$^{H1047R}$ or *PIK3CA*$^{wt}$ were transfected with a Silencer Select siRNA targeting to *TGFA* (ID: s14053) or negative control siRNAs (#1 and #2 mixed in ratio 1:1; all siRNA oligonucleotides from Thermo Fisher Scientific) as previously described (*Pulkkinen et al., 2021*). 48 hr post-transfection, HUVECs were trypsinized, suspended in endothelial basal medium supplemented with 1% FBS, mixed with genetically normal HPF-c (HUVEC-to-HPF-c ratio 8:1), and seeded on 24-well plates at a total density of 15,000 cells/cm$^2$ (i.e. 25,000 HUVECs and 3750 HPF-c/well). Cellular growth in the presence of no additional growth factors was monitored using the IncuCyte S3 Live-cell Imaging System (Essen BioSciences Ltd., Hertfordshire, UK). Images were acquired in 3-hr intervals, four images/well, for a 48-hr period using a 10 x objective. Mean confluency of the cells at each time point was analyzed using the Incucyte software. To calculate the fold change in cell confluency, the mean confluency of each time point was divided by the mean confluency detected at 0 hr timepoint (i.e. after mixing the cells together). Data analysis was finalized by quantitating the relative growth rate in each condition based on a slope of the growth curve (*Figure 4—figure supplement 2*).

## Immunohistology and whole immunomount stainings

Avidin-biotin-HRP system (Vector Laboratories, Burlingame, CA, USA) with 3′–5′-diaminobenzidine (DAP; Zymed, S. San Francisco, CA, USA) color substrate was used for immunohistochemistry on 4–5 µm thick 4% PFA-fixed paraffin-embedded sections. Hematoxylin (Vector Laboratories) was used as a background color. Frozen tissue sections (20–30 µm thick) were fixed for double immunofluorescence staining and blocked with a mixture of 1% BSA and 10% normal goat serum or with 3% normal goat serum. Sections were incubated with primary antibodies and Alexa Fluor 488 or Alexa Fluor 594-conjugated secondary antibodies (A11020, A11037; Thermo Fisher Scientific, dilution 1:200). Mounting was performed with Vectashield medium with DAPI (Vector Laboratories). Sections without primary antibodies were used as negative controls. Primary antibodies for all stainings were as

follows: rabbit anti-TGFA (HPA042297, Sigma-Aldrich, dilution 1:50 and 1:100), rabbit polyclonal anti-pEGFR clone Tyr845 (07–820, Merck, Kenilworth, NJ, USA, dilution 1:50), rabbit polyclonal anti-EGFR ab (HPA018530, Sigma-Aldrich, dilution 1:100), mouse monoclonal CD31 anti-human clone JC70A (M0823, Agilent Dako, Santa Clara, CA, USA, dilution 1:20 or 1:100) and rabbit polyclonal anti-CD31 ab (NB100-2284, Novus Biologicals, Centennial, CO, USA, dilution 1:50). Imaging was performed by Nikon Eclipse Ni-U microscope (10×/0.3 Plan Fluor or 20×/0.5 Plan Fluor objectives; Nikon, Tokyo, Japan) or by Zeiss LSM800 confocal laser scanning microscope using 405/488/561 nm diode lasers together with the appropriate emission filters (20×/0.8 Plan Apochromat, 512 × 512 or 1024x1024 frame size). Maximum intensity projections were generated using the ImageJ program.

## SDS-PAGE electrophoresis and western blot

Cells treated indicated times with rhTGFA (50 ng/ml) were washed with ice-cold PBS, followed by treatment with lysis buffer [50 mM Tris, pH 7.5, 150 mM NaCl, 1 mM EDTA, 1% Triton X-100, 0.5% sodium deoxycholate, 0.1% SDS, 10% glycerol, 1 mM sodium orthovanadate (Sigma-Aldrich), with protease inhibitors (Roche, Basel, Switzerland)]. Equal amounts of total protein (20 μg) from each sample were loaded on the gel and used for analysis on SDS-PAGE electrophoresis and western blot. Primary antibodies used for the immunodetection were phospho-EGFR ab (2234, CST, MA; dilution 1:1000), total EGFR ab (2646, CST, dilution 1:1000), aSMA (M0851, Dako, dilution 1:250), CD31 (M0823, Dako, dilution 1:500), and Vimentin (M0725, Dako, dilution 1:1000). Horse radish peroxidase (HRP)-conjugated secondary antibodies were purchased from Pierce. Antigen-antibody complexes were detected with PIERCE ECL Western Blotting Substrate (Thermo Fisher Scientific) and Gel Dox XR +Gel Documentation System (Bio-Rad Laboratories).

## Mutational analysis

DNA isolation and droplet digital PCR (ddPCR) were performed as previously described (*Nikolaev et al., 2018*). Briefly, DNA isolations from tissue samples were done by lysing 50–100 mg sections of frozen tissue in Hard tissue homogenizing CK28 tubes containing 2.8 mm ceramic beads (Bertin Technologies, Montigny-le-Bretonneux, France) with Precellys homogenizer (Bertin Instruments). Lysed tissues were treated with Proteinase K (Thermo Fisher Scientific) o/n at 50 °C, followed by a DNA extraction with phenol:chloroform:isoamyl alcohol 25:24:1 (Amresco-inc, Solon, OH). Detection of *PIK3CA c.3140A>G* (p.H1047R), *PIK3CA c.3140A>T* (p.H1047L) *PIK3CA c.1633G>A* (p.E545K) and *PIK3CA c.1624G>A* (p.E542K) point mutations were performed on the QX200 ddPCR system (Bio-Rad Laboratories) by using PrimePCR ddPCR Mutation assays according to manufacturer's instructions (Bio-Rad Laboratories). Detection of *TEK c.2740C>T* (p.L914F) mutation was performed by using custom-design Taqman SNP Genotyping assays [Thermo Fisher Scientific; fwd 5'-CTTCCCTCCAGG CTACTT-3', rev 5'-AATGCTGGGTCCGTCT-3', reporter 1 (HEX) 5'-CTTGCGAAGGAAGTCCAGAA GGTTTC-3', and reporter 2 (FAM) 5'- CTTGCGAAAGAAGTCCAGAAGGTTTC-3']. Synthetic construct gBlocks Gene Fragments (IDT, Coralville, Iowa, USA; *Supplementary file 2*) with and without a mutation were designed for each assay and used as positive control DNAs in ddPCR. DNA samples with mutation-positive event >10 and fractional abundance >0.5% were considered as mutation positive.

## A modified xenograft model for the vascular lesion

Animal experiments were approved by the National Experimental Animal Board of Finland (Decision No Esavi-2019–004672) and carried out in accordance with the guidelines of the Finnish Act on Animal Experimentation. $2.5 \times 10^6$ HUVECs expressing *PIK3CA*^wt or *PIK3CA*^H1047R were suspended in growth factor-reduced and phenol red-free Matrigel (Corning, New York, USA) with or without $0.8 \times 10^6$ HPF-c cells and injected s.c. into both flanks of 6-weeks-old female Athymic *Nude-Foxn1*^nu mice (n=18; Envigo, Indiana, USA). For comparison lesion growth with or without the oncogenic PIK3CA variant, each mouse had one plug with *PIK3CA*^H1047R ECs and one with *PIK3CA*^wt ECs. Prior to injections, mice were randomized to groups receiving either ECs or ECs+ HPF cs, or to be treated with or without afatinib. Lesion size was measured twice a week from d4 onwards with a digital caliper. After lesions reached ~200 μm³ in size, 25 mg/kg afatinib was given to mice once daily p.o. for 9 days (diluted in 10% DMSO, 40% PEG300, 5% Tween-80, and 45% saline, MedChemExpress LLC, NJ). Lesions were dissected at d18-20, and lesion size was measured from the dissected explants (NIS-Elements AR). Volume was calculated with the formula volume = (length × width²)/2, where the length

is the longest diameter and the width is the shortest diameter of the lesion. Explants were fixed with 4% paraformaldehyde for 4 hr and embedded in paraffin. Vascularization, amount of erythrocytes, and overall inflammation were evaluated from hematoxylin and eosin (H&E) or CD31 staining by visual inspection in a blinded manner by one observer (H&E) or two independent observers (CD31) and scored on a scale of 0–3, 0 being the lowest score and 3 the highest (0 no vascularization, no erythrocytes, no inflammation/ 1 a few vascular channels, a few vascular channels filled with erythrocytes, mild inflammation/ 2 many vascular channels, many vascular channels filled with erythrocytes, moderate inflammation/ 3 a lot of vascular channels, most of the channels filled with erythrocytes, severe inflammation). EGFR expression was scored (0 with no staining, 1 with low amount, 2 with moderate amount, 3 with high amount). Exclusion criteria from the analysis were: (i) unsuccessful plug formation; and (ii) different anatomical locations of the plug. In addition, quantification of relative erythrocyte area (i.e. segmented area containing erythrocytes/total plug area) was performed on H&E-stained sections using an automated intensity-based SproutAngio quantification tool (https://github.com/mbbio/SproutAngio, *Beter et al., 2023*) from whole plug images scanned with an Olympus VS200 slide scanner using a 20×/0.5 Plan Fluor objective.

## Statistical analysis

Results are expressed as means ± SEM. Statistical significance was analyzed using the Kruskal-Wallis test with two-stage step-up method of Benjamini, Krieger, and Yekutieli to control FDR (3 or more groups, data not normally distributed); Brown-Forsythe and Welch ANOVA with Dunnet T3 post hoc test or two-stage step-up method of Benjamini, Krieger and Yekutieli to control FDR (3 or more groups, data normally distributed but unequal variances); One-way ANOVA with Bonferroni post hoc test (3 or more groups, data normally distributed with equal variances); Two-tailed Mann-Whitney U test (2 groups, data not normally distributed); or two-tailed unpaired t-test with Welch's correction (2 groups, data normally distributed but unequal variances). $p < 0.05$ was used to define a significant difference between the groups. Correlation between two markers was analyzed using Spearman rho (data not normally distributed), with values >0.6 showing strong correlation and values 0.4–0.59 moderate correlation.

## Results

### Patient demographics

Patient samples were classified according to ISSVA guidelines by a pathologist specialized in vascular anomalies (*Tables 1–2*). 35 patients were included in the study having VM (n=15) or AST (n=20). Additionally, three patients were classified as VM/AST having characteristic features of both vascular anomalies. All lesions were unifocal, except three multifocal VM lesions. Median age of the AST and VM patients was 18 (range 11–46 years, female-to-male ratio 14:6) and 31 (range 9–77 years female-to-male ratio 6:9), respectively. Most of the AST lesions (85%) are located in the extremities. Of all AST lesions, 60% were intramuscular lesions (12/20), five lesions affected both intramuscular and subcutaneous tissue, and one was both synovial membrane and intramuscular tissue. 60% of the VM lesions were from the extremities. Of all VM lesions, three were intramuscular, two located in both intramuscular and subcutaneous tissue, and two affected synovial membrane, intramuscular, and subcutaneous tissue. All VM/AST lesions (n=3, 100%) were from the extremities, of which one was intramuscular. Genetic mutations were detected from vascular lesions by droplet digital PCR (ddPCR). Oncogenic *PIK3CA* variants were detected in 19/38 patients (AST 75%, VM 18%; *Tables 1–2*), of which *PIK3CA* p.H1047R/L somatic mutation was found in 10/19 patients. 53% of patients had received sclerotherapy (VM 3/15, AST 15/20, VM/AST 2/3 patients, respectively). A representative magnetic resonance image of the AST lesion located in an ankle of a 13-year-old male is presented in *Figure 1A*. A representative 3D confocal image of vessel organization in the AST lesion is shown in *Figure 1B*. CD31-labeled longitudinal vessels were shown to be torturous, branched, and variable in size (*Figure 1B*).

### TGFA is upregulated in patient-derived ECs and vascular lesions with the venous component

Five fresh tissue samples, including four AST and one VM, were obtained for bulk RNA-sequencing experiments. After digestion steps, ECs were selected by CD31 microbead kit. Somatic mutations in

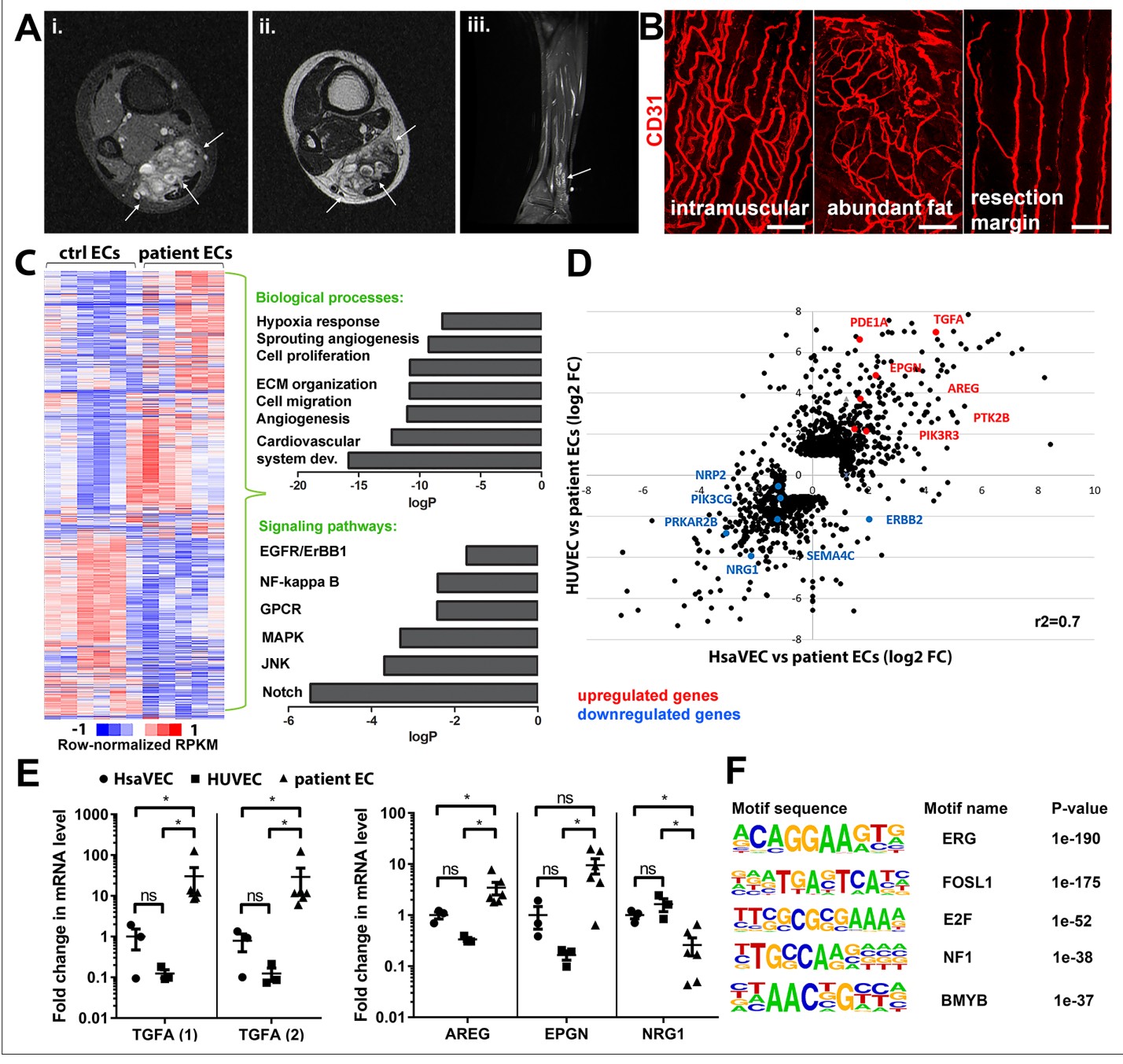

**Figure 1.** Genes involved in ErbB signaling pathway are upregulated in patient-derived endothelial cells (ECs) in vascular lesions with the venous components. (**A**) Magnetic resonance images of an angiomatosis of soft tissue (AST) lesion (arrows) in the soleus muscle show the replacement of the normal muscle by dilated venous channels, diffusely enhancing small vessels and adipose tissue. (i) Axial T1-weighted fat-saturated contrast-enhanced image. (ii) Axial T2-weighted image. (iii) Sagittal T2-weighted fat-saturated image. (**B**) 1mm-thick whole immunomounts were prepared from patient lesions, immunolabeled, and imaged by laser scanning confocal microscopy. Images of the vasculature in the AST lesion located in the shin of a 16-year-old female are shown. Endothelial cells are immunolabeled with CD31 antibody (red). Vasculature of the same lesion in the intramuscular area (**i**), with abundant fat (ii) and next to resection margin (iii) are presented. Longitudinal vessels are seen. Scale bars, 100 μm. (**C**) Heatmap of normalized RPKM values (−1–1) of the differentially regulated genes in patient-derived ECs compared to HUVEC and human saphenous vein endothelial cells (HsaVEC) control cells detected by bulk RNA-seq. Clustering was performed using Spearman's rank correlation. Biological processes and cell signaling pathways detected by gene ontology analysis in patient-derived ECs. (**D**) Scatter plot of the fold changes in gene expression comparing patient-derived and control ECs. Selected genes involved in PIK3CA, VEGFR2, and ErbB1-4 signaling are highlighted in red (upregulated) and blue (downregulated). Pearson correlation value ($r^2$) is shown. (**E**) Changes in mRNA expression levels of ErbB ligands (transforming growth factor A (*TGFA*), with two different assays; amphiregulin, *AREG*; epigen, *EPGN1*; neuregulin 1, *NRG1*) were validated with RT-qPCR from patient-derived and control ECs. Mean and SEM

*Figure 1 continued on next page*

Figure 1 continued

is presented (HsaVEC and HUVEC, n=3; patient ECs, n=5). Kruskal-Wallis test with two-stage step-up method of Benjamini, Krieger, and Yekutieli to control FDR [*TGFA* (1), *TGFA* (2), *AREG*; data not normally distributed] or Brown-Forsythe and Welch ANOVA with Dunnet T3 post hoc test (*NRG1*, *EPGN*; data normally distributed but unequal variances). *p<0.05. (**F**) Sequence motifs associated with differentially regulated genes in patient-derived ECs.

The online version of this article includes the following figure supplement(s) for figure 1:

**Figure supplement 1.** Highest expression of TGFA and EPGN is shown in patient ECs positive for oncogenic PIK3CA variants.

**Figure supplement 2.** TGFA and genes related to cell migration, proliferation, and ECM organization, are regulated in patient ECs with oncogenic PIK3CA variant.

*PIK3CA* were detected by ddPCR in lesions from 3 out of 5 patients (fractional abundance 44–51%, *Tables 1–2*), and for the first time, in ECs isolated from AST lesions (*Table 1*). Bulk RNA-sequencing was used to compare gene expression profiles of patient-derived [CD31+]ECs to healthy ECs derived from umbilical cords (HUVEC) or saphenous veins (HsaVEC). With principal component analysis, VM was not distinguished from AST samples and thus, was kept in the analysis. 1128 and 571 genes were found to be differentially expressed between control cells and patient-derived [CD31+]ECs, respectively (*Figure 1C–D*). Differentially expressed genes (DEGs) were involved in angiogenic cellular processes, such as cell proliferation, migration, extracellular matrix organization, and hypoxia (*Figure 1C*; *Supplementary file 1*). Of particular interest were the fourteen genes found to be involved in ErbB signaling pathway known to regulate pathological angiogenesis. Multiple ligands of ErbB1-4 receptors were detected, e.g., transforming growth factor A (*TGFA*), amphiregulin (*AREG*), neuregulin-1, and epigen (*EPGN*). Also, G protein-coupled receptor signaling and RAS/MAPK cascade were found to be regulated (*Figure 1C*; *Table 3*), previously linked to ErbB activation and downstream signaling (*Wee and Wang, 2017*; *Thomas et al., 2006*). Similar genes and signaling pathways, e.g., *TGFA* and cell migration, proliferation, and ECM organization, were shown to be regulated in a separate analysis done for patient ECs with oncogenic *PIK3CA* variant only in comparison to control ECs (*Figure 1—figure supplements 1 and 2*).

Significant upregulation of ErbB1/EGFR ligands *TGFA* and amphiregulin (*AREG*) was validated by RT-qPCR in patient-derived[CD31+]ECs in comparison to control ECs, whereas no difference was observed in the regulation of ErbB4 ligand *EPGN* (*Figure 1E*; *De Luca et al., 2008*; *Ginsberg, 2007*). In accordance with data from bulk RNA-sequencing, ErbB2-ErbB4 binding neuregulin-1 was downregulated in patient-derived [CD31+]ECs (*Figure 1E*). De novo motif analysis of regulatory regions, i.e., enhancers within 100 kb of the gene transcriptional start site, was further used to identify possible regulatory transcription factor binding sites in patient-derived ECs that could regulate their phenotype. Cell cycle regulators E2F and FOSL1 were found to be the major regulators of transcription activity together with EC-specific transcription factor ERG (*Figure 1F*).

Next, expressions of TGFA and AREG were validated at the tissue level from patient lesions. *TGFA* mRNA was shown to be upregulated in both AST and VM by RT-qPCR (*Figure 2A*, *Figure 2—figure supplement 1*, n=8 VM, n=3 AST) in comparison to control tissue, whereas no change of *AREG* mRNA was observed (*Figure 2—figure supplement 2*). By immunohistochemistry, TGFA was detected in the endothelium, pericytes, and intervascular stromal cells in both AST and VM lesions (*Figure 2B–C*; *Figure 2—figure supplement 2*; positivity in 4/5 VM, 9/10 AST, 1/1 VM/AST lesions). Activated EGFR pathway was further demonstrated in AST lesions by detecting phosphorylated EGFR (positivity in 9/9 AST lesions; *Figure 2D*), with most of the signal located in intervascular stromal cells. A heatmap for scoring of TGFA and pEGFR expression levels and the presence of an oncogenic PIK3CA variant is presented in *Figure 2E*, showing moderate or strong expression of these factors in the lesions.

In support of findings in immunohistochemistry, *TGFA* mRNA upregulation or secretion was detected in patient-derived [CD31+]ECs and in intervascular stromal[CD31-, vimentin+] cells (Patient SCs; *Figure 2F–G*). By morphology patient SCs resembled fibroblasts, and (*Figure 2F*) were characterized by western blot to be negative for CD31 marker and positive for fibroblast and/or smooth muscle cell marker vimentin (*Figure 2—figure supplement 2*). None of the patient SCs had *PIK3CA* mutations detected by ddPCR (*Table 1*). Higher expression of *EGFR* mRNA was detected in control fibroblasts and patient SCs in comparison to ECs (*Figure 2—figure supplement 2*). The data was in-line with scRNAseq data from mouse lower limb skeletal muscle (Tabula Muris, czbiohub.org) where *Egfr* did

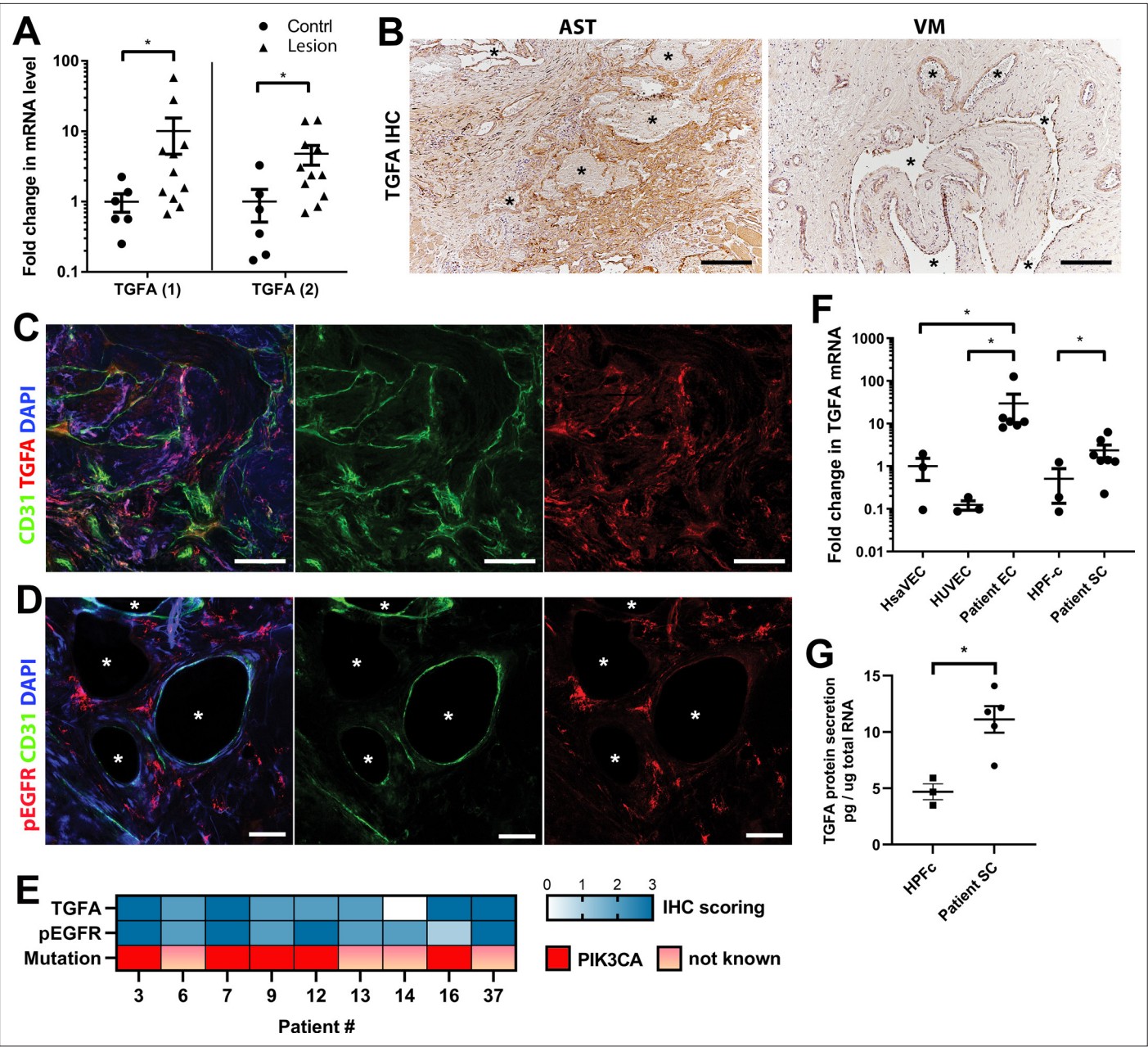

**Figure 2.** Epidermal growth factor receptor (EGFR)/ErbB1 ligand transforming growth factor A (TGFA) is upregulated in venous malformation (VM) and angiomatosis of soft tissue (AST) patient tissue samples. (**A**) RT-qPCR analysis showed significantly higher expression of *TGFA* mRNA (with two different assays) in VM and AST tissue than in the control group. Mean and SEM are presented (lesions n=11; control group, n=6), analyzed using two-tailed Mann-Whitney U test. *, p<0.05. (**B**) Representative images of TGFA expression in AST and VM patient samples. See *Figure 2—figure supplement 2* for normal skeletal muscle control. TGFA signal was detected in lesions in the endothelium, pericytes, and intrastromal cells by immunohistochemistry. Asterisks point out the largest vascular lumens which, especially in AST, are commonly tightly packed with erythrocytes. Scale bars, 200 µm. (**C**) Representative whole immunomount images of CD31-labeled endothelium (green) and TGFA expression (red) in a patient diagnosed with intramuscular AST. Nuclei are stained with DAPI (blue). Longitudinal vessels are seen. Scale bars, 100 µm. (**D**) Representative confocal images of phosphorylated epidermal growth factor receptor (EGFR) (red) expression in AST lesion. Endothelium is labeled with CD31 antibody (green), and nuclei with DAPI. Vascular lumens are indicated with white asterisks in the cross-sections. Scale bars, 50 µm. (**E**) Heatmap of TGFA and pEGFR protein expression and presence of oncogenic PIK3CA variants in AST patients. Level of protein expression was scored (0–3) based on the detected signal in immunocytochemistry (0, none; 1, low; 2, medium; and 3, high). (**F**) RT-qPCR analyses of *TGFA* expression in patient-derived endothelial cells (ECs) and intervascular stromal cells (SCs). Selection of ECs was performed by CD31 MicroBead Kit. Stromal cells were characterized by western blot and showed to be negative for EC marker CD31, and positive for fibroblast and smooth muscle cell marker vimentin (see *Figure 2—figure supplement 2*). The data is presented as relative mean fold change to HsaVEC control group and SEM (Human saphenous vein endothelial cells:HsaVEC, HUVEC, and HPF-c,

*Figure 2 continued on next page*

*Figure 2 continued*

n=3; patient ECs, n=5; patient SCs, n=6. Statistical significance was analyzed using Kruskal-Wallis test with two-stage step-up method of Benjamini, Krieger and Yekutieli to control FDR (ECs) or two-tailed Mann-Whitney U test (SCs; in both analyses, data not normally distributed). *p<0.05. (**G**) TGFA was shown to be secreted from patient-derived intervascular stromal cells (SCs) by ELISA (patient SC n=5; HPF-c n=3). Two-tailed unpaired t-test with Welch's correction (data normally distributed but unequal variances). **p<0.005.

The online version of this article includes the following source data and figure supplement(s) for figure 2:

**Figure supplement 1.** TGFA expression in patient lesions normalized to vascular EC marker.

**Figure supplement 1—source data 1.** Raw data for western blot images.

**Figure supplement 2.** TGFA and EGFR are expressed in patient cells.

**Figure supplement 3.** scRNAseq data from mouse lower limb skeletal muscle presenting clusters positive for *EGFR* and *TGFA*.

express in mesenchymal stem cells and skeletal muscle satellite cells but only in a small portion of ECs (***Figure 2—figure supplement 3A–B***). On the contrary, a small number of *Tgfa*+ cells were detected in mouse normal healthy skeletal muscle, showing the highest number of *Tgfa*+ cells in the EC cluster (***Figure 2—figure supplement 3C–D***).

Altogether, these results suggest that ErbB binding ligands are upregulated in AST and VM, and that TGFA, demonstrated to induce angiogenesis in earlier studies (***De Luca et al., 2008***; ***Leker et al., 2009***; ***Schreiber et al., 1986***) could be among the potential factors to induce a pro-angiogenic phenotype of lesion ECs.

## Oncogenic PIK3CA p.H1047R induces expression of TGFA and enrichment of hallmark hypoxia

To understand the mechanism behind TGFA upregulation in patient lesions, and the potential role of the oncogenic *PIK3CA* variant in it, bulk RNA-sequencing was performed on lentiviral-transduced ECs that expressed either wild-type or oncogenic *PIK3CA* p.H1047R, the most common somatic mutation found from this patient cohort (***Tables 1–2***; ***Figure 3***; ***Figure 3—figure supplements 1 and 2***). In-line with our experiments with patient-derived [CD31+]ECs, *TGFA* mRNA expression was shown to be induced in *PIK3CA*[H1047R] expressing ECs (***Figure 3—figure supplement 1D***). In addition, GO analysis of these cells showed a hallmark 'mTORC1 signaling,' indicative of activation of the signaling pathway downstream from PIK3CA, as well as hallmarks 'Glycolysis,' further indicative of a metabolic change from normal oxygen consumption towards anaerobic energy metabolism (***Figure 3A***). Interestingly, despite the normoxic cell culture conditions, hallmark hypoxia was detected as one of the top enriched hallmarks in *PIK3CA*[H1047R] expressing ECs by GO analysis (***Figure 3A***). Further comparison to the RNA-sequencing data from ECs expressing constitutively active hypoxia-inducible factors (*HIF*s; GSE98060) confirmed that among 47 independent DEGs detected from our patient-derived [CD31+]ECs (***Figure 1***) under 'Response to hypoxia' (GO: 0001666) or 'Hallmark Hypoxia,' the majority (28 DEGs) were significantly altered by *HIF1A*, *HIF2A* or oncogenic *PIK3CA*[H1047R]. This indicated that part of the HIF-regulated genes were also direct transcriptional targets downstream of the oncogenic PIK3CA signaling. Top 15 hypoxia-related DEGs are presented in ***Figure 3B***.

As HIFs and their target genes have previously been shown to induce angiogenesis or expression of TGFA and VEGF-A (***Zong et al., 2017***; ***Gunaratnam et al., 2003***; ***Lee et al., 2018***; ***Krishnamachary et al., 2003***) and depletion of HIF1A/HIF2A has been shown to lead to downregulation in TGFA expression (***Chang et al., 2013***) correlation of these factors was next studied in lesions. The data showed a strong positive correlation between *TGFA* and *HIF1A* mRNA expression levels ($\rho$ =0.73), and a moderate correlation between *TGFA* and *HIF2A* ($\rho$ =0.45) (***Figure 3C–D***, n=12 AST, n=7 VM, n=2 VM/AST). Additionally, a positive correlation ($\rho$ =0.62) between *VEGFA* and *TGFA* mRNA expression levels (***Figure 3E***) was observed. VEGF-A expression or secretion was further validated by immunohistochemistry in patient lesions (***Figure 3F–G***) or in patient SCs by RT-qPCR and ELISA (***Figure 3—figure supplement 3***).

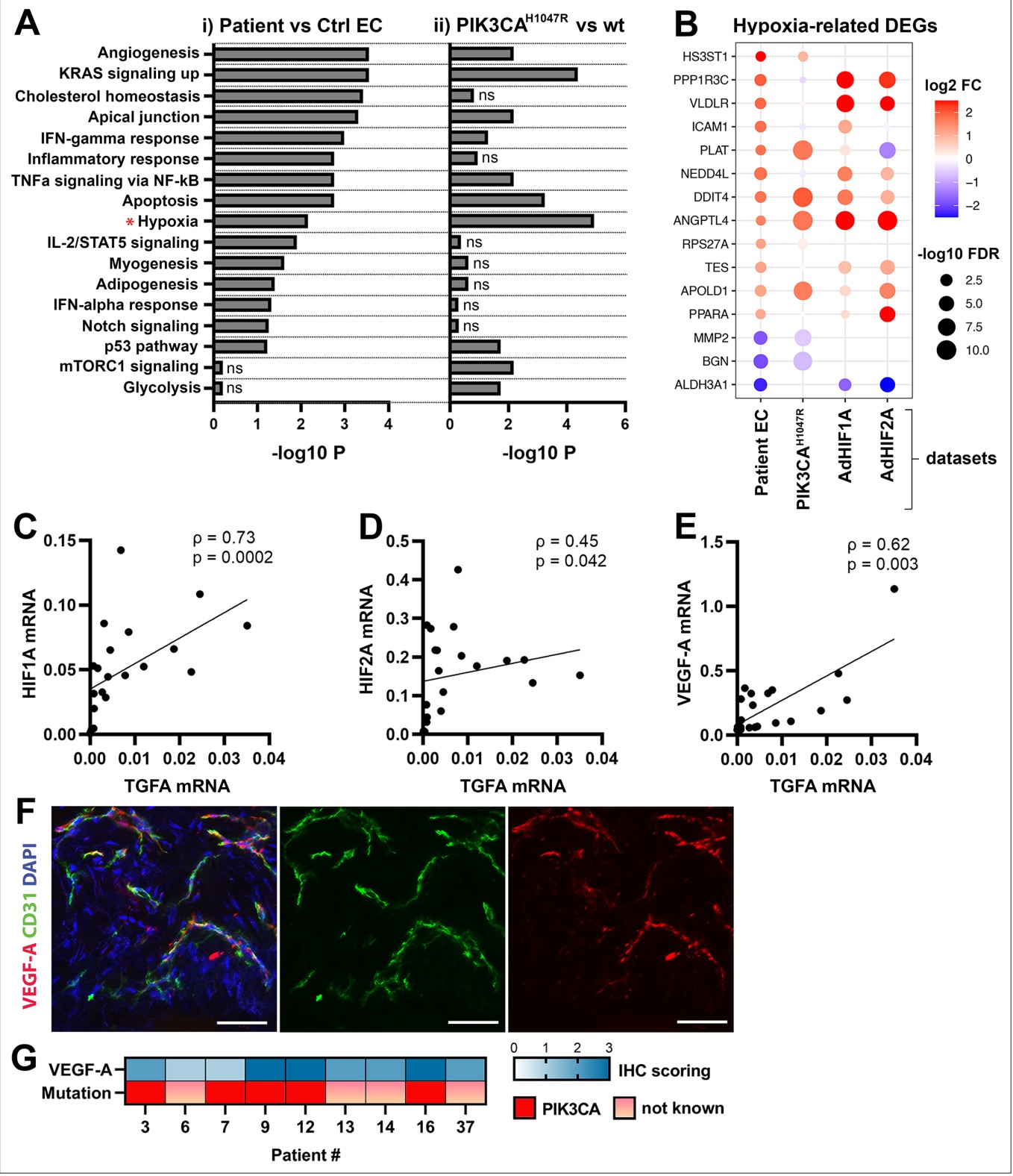

**Figure 3.** Oncogenic *PIK3CA*[H1047R] induces enrichment of hallmark hypoxia. (**A**) Several shared MSigDB Hallmarks were found in bulk RNA-seq data from (i) patient-derived endothelial cells (ECs) vs control ECs (left panel), and (ii) ECs expressing *PIK3CA*[wt] or most common oncogenic variant detected in patient lesions, *PIK3CA*[H1047R] (right panel). Hallmark analysis was performed with the EnrichR web server, using adjusted p-value <0.1 to define terms with significant enrichment of differentially expressed genes (DEGs). Hallmark Hypoxia (*) was detected as one of the top hallmarks in both RNA-seq datasets. (**B**) Top 15 hypoxia-related genes differentially expressed between patient-derived vs control ECs, shown to be regulated in *PIK3CA*[H1047R]

*Figure 3 continued on next page*

*Figure 3 continued*

expressing ECs and/or by *HIF1A/HIF2A*. (**C–E**) Correlation between transforming growth factor A (*TGFA*) and *HIF1A* (**C**), *HIF2A* (**D**), and *VEGFA* (**E**) mRNA expression levels detected in angiomatosis of soft tissue (AST) and venous malformation (VM) lesions (n=23). Spearman's test was used to define correlation (data not normally distributed). Rho, Spearman's rank correlation coefficient. (**F**) Representative whole immunomount images of vasculature in AST lesion expressing VEGF-A (red) detected by confocal microscopy. Endothelium is labeled with CD31 antibody (green) and nuclei with DAPI (blue). Co-localization of VEGF-A and CD31 markers are seen in yellow. Longitudinal vessels are seen. Scale bars, 50 μm. (**G**) Heatmap of VEGF-A expression and presence of oncogenic *PIK3CA* variants in AST patients. Level of VEGF-A expression was scored (0–3) based on the detected signal in immunocytochemistry (0, none; 1, low; 2, medium; and 3, high).

The online version of this article includes the following figure supplement(s) for figure 3:

**Figure supplement 1.** Oncogenic *PIK3CA*[H1047R] induces morphological and transcriptional changes in transduced ECs.

**Figure supplement 2.** Comparison of bulk RNA-sequencing data of patient-derived endothelial cells and *PIK3CA*[H1047R] transduced primary endothelial cells.

**Figure supplement 3.** VEGF-A is upregulated and secreted from patient-derived stromal cells.

## Patient SCs and TGFA induce a pro-angiogenic EC phenotype together with VEGF-A

To study further the effect of VEGF/TGFA-secreting patient SCs on ECs, we used a modified fibrin bead angiogenesis assay (*Nakatsu et al., 2007*). Patient SCs were shown to induce HUVEC sprouting without any additional growth factor stimulation (*Figure 4A–B*), implying that paracrine factors secreted by the SCs modulate the EC phenotype. We hypothesized that the mechanism could be via TGFA-mediated upregulation of VEGF-A. Accordingly, a significant increase in *VEGFA* mRNA and protein secretion was observed after stimulation of control fibroblasts (HPF-c) with rhTGFA (*Figure 4—figure supplement 1*). Also, in a fibrin bead assay with HUVECs, co-stimulation with rhVEGF-A and rhTGFA proteins resulted in a higher increase of EC area and sprouting in comparison to either of the growth factors alone (*Figure 4C–E*). This suggested that these growth factors have a synergistic effect on modulating EC phenotype.

In contrast to TGFA (*Figure 3—figure supplement 1D*), no change in *VEGFA* mRNA expression was seen in *PIK3CA*[H1047R] expressing ECs. Neither did rhTGFA significantly affect cell proliferation in cultures with ECs or HPF-c alone. To understand the importance of TGFA expression in the growth of ECs in the presence of *PIK3CA*[H1047R] mutation and HPF-c or SC, proliferation assays were performed in co-culture conditions. Co-culturing of *PIK3CA*[H1047R] expressing ECs together with HPF-c showed an increased growth rate in comparison to *PIK3CA*[wt]-treated cells. Importantly, the response was abolished when TGFA expression in ECs was knocked down by siRNA (*Figure 4F*; *Figure 4—figure supplement 2*). Thus, altogether the data suggests that TGFA expression, induced by oncogenic *PIK3CA*[H1047R], results in a pro-angiogenic EC phenotype by increasing cell proliferation and VEGF-A secretion but only in conditions where ECs and HPF-c/SCs are combined.

## Fibroblasts induce vascularization in a mouse xenograft model for vascular lesions

Due to the small number of cells obtained from patient lesions, further studies to understand the role of SCs/fibroblasts in lesion formation in vivo were performed with commercially available primary cells. A new modified mouse xenograft model based on *Goines et al., 2018* was used for the first time combining: (i) ECs expressing either oncogenic *PIK3CA*[H1047R] or *PIK3CA*[wt], and (ii) genotypically normal primary fibroblasts (HPF-c; *Figure 5A*). Lesion growth and size at d20 was observed to be similar between *PIK3CA*[H1047R]-transduced ECs with or without fibroblasts (*Figure 5B*). With H&E staining, various sized vascular channels filled with erythrocytes were detected (*Figure 5D*). Notably, there was very little if any blood-filled vascularization shown in explants with ECs expressing *PIK3CA*[wt], whereas vascular channels in explants with *PIK3CA*[wt] expressing ECs and fibroblasts clearly contained erythrocytes (*Figure 5C–D H*). Higher vascularization, detected with EC marker CD31, was observed with explants with embedded fibroblasts in comparison to ECs alone (*Figure 5E–F*). In comparison to explants containing only *PIK3CA*[H1047R] ECs, the vascular channels with fibroblasts were wider and appeared more irregular in shape (*Figure 5D–F*). The highest CD31 vascularization score

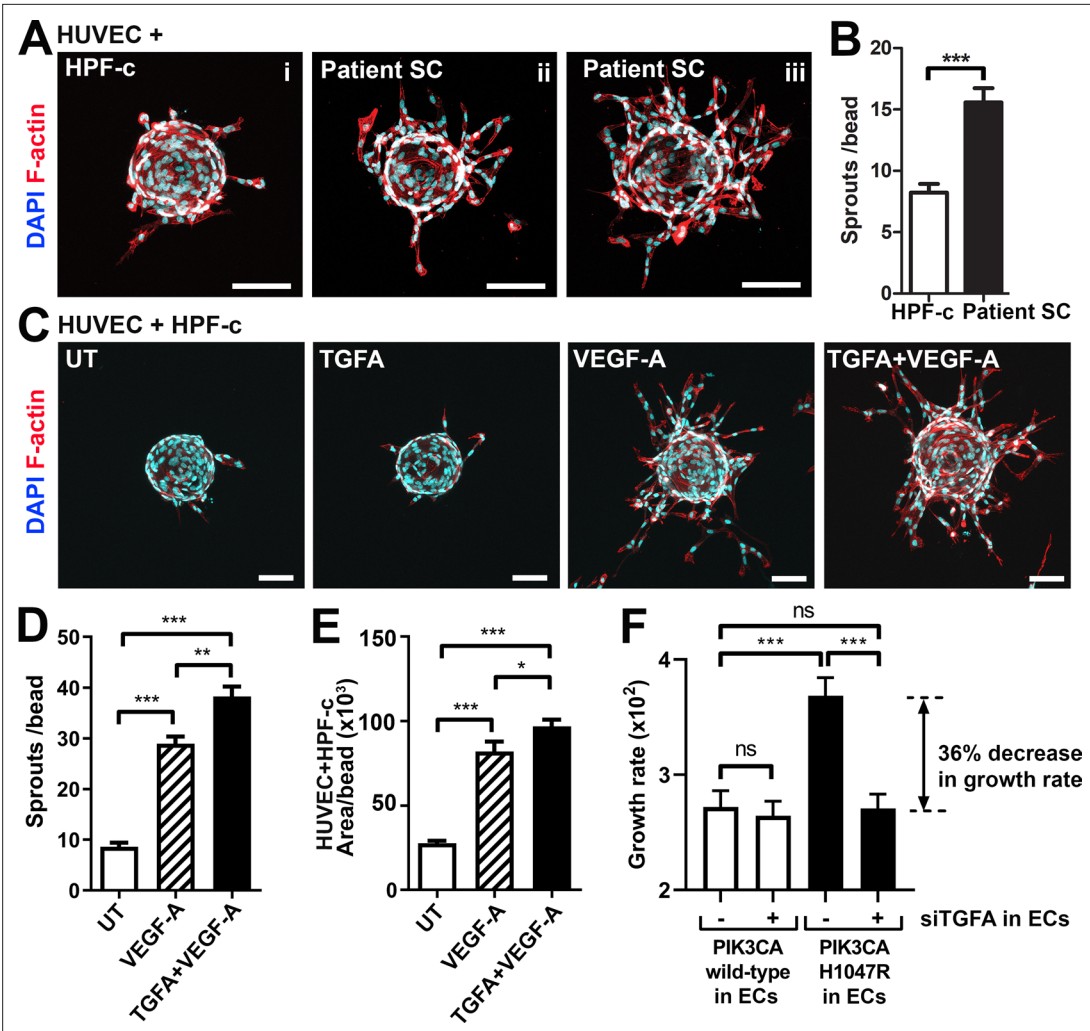

**Figure 4.** Patient stromal cells (SCs) and transforming growth factor A (TGFA) induce an angiogenic endothelial cell (EC) phenotype together with VEGF-A. (**A–B**) Venous malformation (VM) patient SCs induce sprouting of genotypically normal ECs. HUVECs on collagen-coated beads were embedded into a fibrin gel and patient SCs or control HPF-c cells were put on top. Representative images are presented at d7. ECs are labeled with phalloidin (red). nuclei with DAPI (blue; **A**) The number of sprouts per bead in each condition is shown (**B**). Two independent experiments were done in triplicates. Two-tailed Mann-Whitney U test (data not normally distributed). *p<0.05. In all images, scale bar is 100 µm. (**C–E**) Fibrin bead assay with HUVECs and HPF-c cells shows increased EC sprouting after stimulation with rhVEGF-A and rhTGFA at d6. ECs are labeled with phalloidin (red), and nuclei with DAPI (blue; **C**). The number of sprouts per bead (**D**) or sprout area (**E**) in each condition was determined from confocal images by ImageJ (45 beads/group). 2 independent experiments were done in triplicates. Brown-Forsythe and Welch ANOVA with Dunnet T3 post-hoc test (**D**) or Kruskal-Wallis test with two-stage step-up method of Benjamini, Krieger and Yekutieli to control FDR (**E**). ***p<0.001. (**F**) Co-culture experiments with HPF-c and HUVEC cells showed an increased growth rate in wells with $PIK3CA^{H1047R}$-expressing ECs in comparison to wells with ECs expressing $PIK3CA^{wt}$. The response was abolished after inhibition of endogenous *TGFA* in ECs by specific siRNA, demonstrating the involvement of TGFA in $PIK3CA^{H1047R}$-induced responses. Wells with siCtrl-transduced ECs (marked to be negative for *siTGFA*) were used as a control group in the experiments. Cellular growth was monitored using IncuCyte Live-Cell Imaging system. Data are presented as relative growth rate from two experiments done in triplicates. One-way ANOVA with Bonferroni or Sidac post-hoc test. ***p<0.001. In all data, mean and SEM are presented.

The online version of this article includes the following figure supplement(s) for figure 4:

**Figure supplement 1.** rhTGFA stimulation induces VEGFA in normal fibroblasts.

**Figure supplement 2.** Quantification for the proliferation of cells in co-culture experiments combining endothelial cells (ECs) expressing $PI3KCA^{H1047R}$ or $PI3KCAwt^{wt}$+/−*TGFA* and genetically normal HPF-c.

and erythrocyte amount were detected in explants with ECs expressing $PIK3CA^{H1047R}$ and fibroblasts (*Figure 5G–H*). In addition, a higher number of inflammatory cells was seen in the explants with ECs expressing $PIK3CA^{H1047R}$ than $PIK3CA^{wt}$; however, no statistical difference was detected with or without fibroblasts (*Figure 5*).

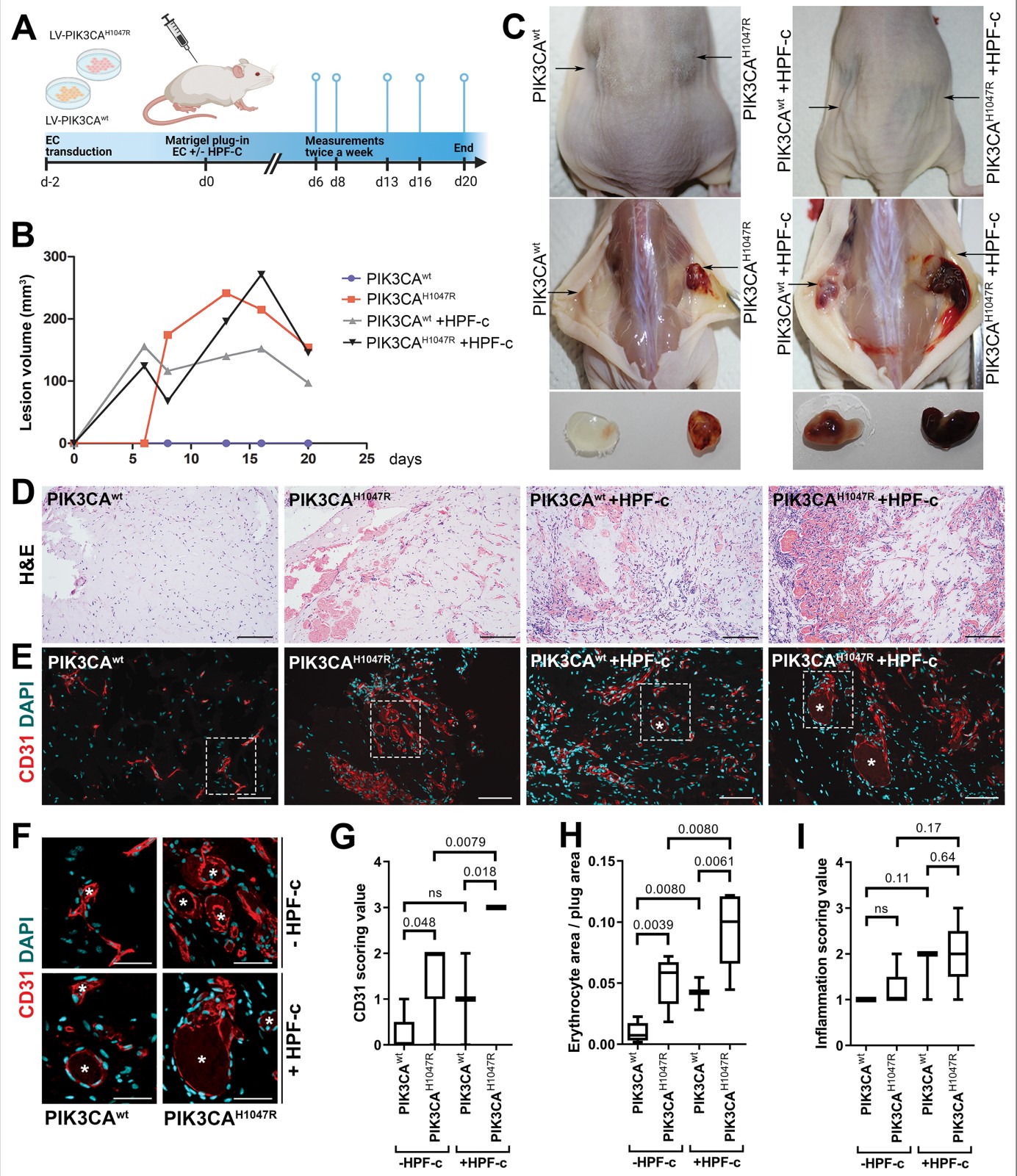

**Figure 5.** Fibroblasts induce vascularization in a mouse xenograft model for vascular lesions. (**A**) Subcutaneous injection of matrigel with HUVECs transduced with *PIK3CA*^wt or *PIK3CA*^H1047R encoding lentivirus vectors, with or without primary fibroblasts, was performed in athymic *Nude-Foxn1*^nu mice. A timeline of the animal experiment is presented. (**B**) Lesion volume measured by caliper from day 6 to day 20 (n=5 EC^wt, n=5 EC^H1047R, n=3 EC^wt+ FB, n=5 EC^H1047R+ FB). (**C**) Representative images of mice and dissected lesion explants on day 20. (**D, E**) Explant sections stained with hematoxylin

*Figure 5 continued on next page*

*Figure 5 continued*

and eosin (H&E; **D**) or EC marker CD31 (red) and DAPI (nuclei, blue; **E**). Scale bars, 200 μm, H&E; 100 μm, CD31. (**F**) Close-up of the vascular lumens detected in the explants with or without fibroblasts (CD31, red; DAPI, blue). Scale bars, 50 μm. (**G**) Scoring for vascularization of CD31-stained sections. The highest vascularization score was observed in the explants with HUVECs expressing PIK3CA[H1047R] and fibroblasts. (**H, I**) Quantification of erythrocyte area/plug (**H**) and inflammation scoring were (**I**) done on H&E-stained sections. Data are presented as mean and variation, and analyzed using (**G, I**) two-tailed Mann-Whitney U test (to compare scoring between each 2 groups) or (**H**) Brown-Forsythe and Welch ANOVA with two-stage step-up method of Benjamini, Krieger and Yekutieli to control FDR. Exact p-values are presented and *p<0.05 considered as statistically significant difference.

The online version of this article includes the following figure supplement(s) for figure 5:

**Figure supplement 1.** EGFR expression in mouse xenograft sections.

Further immunohistochemistry confirmed the production of EGFR protein in the explants with *PIK3CA*[H1047R] -expressing ECs; however, no difference was detected between explants with or without fibroblasts (*Figure 5—figure supplement 1*). In summary, the data indicates the importance/potential of fibroblasts in inducing aberrant vasculature in lesions.

## ErbB family antagonist Afatinib reduces VEGF secretion, angiogenic sprouting, and lesion size

Next, afatinib (Gilotrif, Giotrif), an inhibitor of EGFR/ErbB1, ErbB2, and ErbB4, was used to test the potential inhibition of EC angiogenesis and vascular lesion growth in mice. First, in vitro, afatinib was shown to block TGFA-stimulated EGFR phosphorylation detected by western blot (*Figure 6A*)**,** and to decrease VEGF-A secretion from TGFA-stimulated control fibroblasts measured by ELISA (*Figure 6B*). Accordingly, in the fibrin bead assay, afatinib was shown to reduce rhVEGF-A and rhTGF-A mediated EC sprouting (*Figure 6C–D*). To further test the effect of afatinib on PI3K-driven vascular lesion

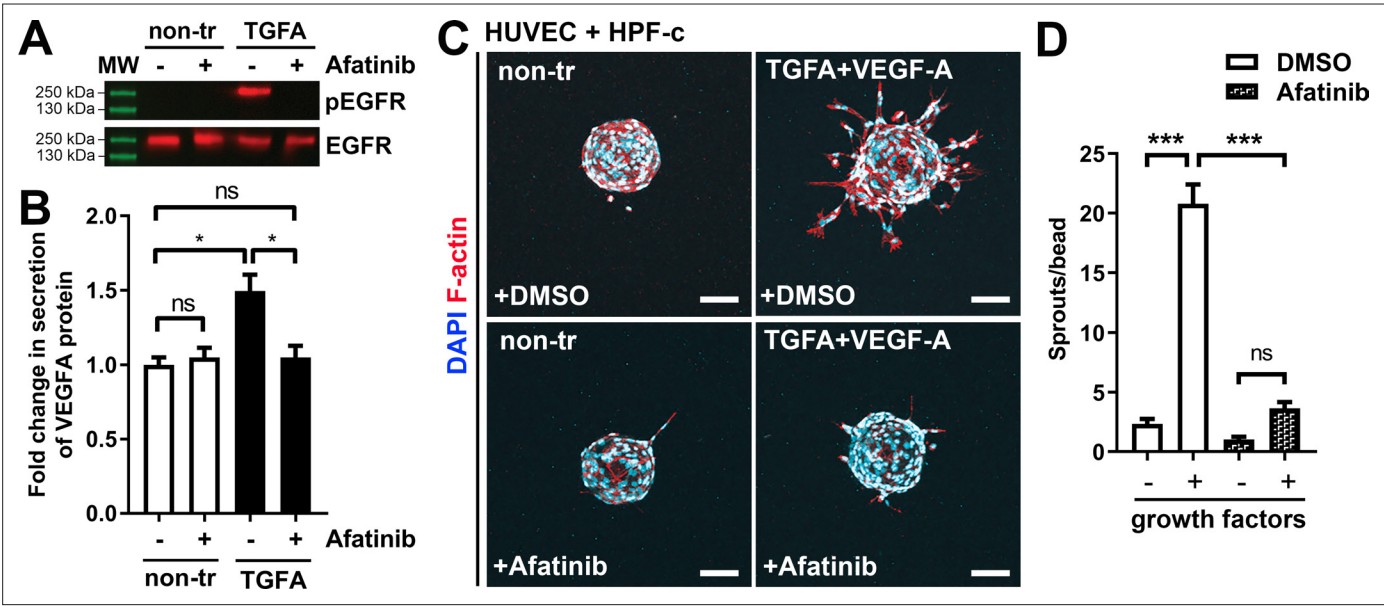

**Figure 6.** Afatinib reduces VEGF-A secretion, angiogenesis, and lesion size. (**A**) Afatinib decreased EGFR/ErbB1-phosphorylation in rhTGFA-stimulated HPF-c cells. Total epidermal growth factor receptor (EGFR) was used to control equal loading of the samples. (**B**) VEGF secretion measured by ELISA from rhTGFA-stimulated control fibroblasts (HPF-c) with or without afatinib treatment. Two independent experiments done in triplicates are presented as mean and SEM. Brown-Forsythe and Welch ANOVA with Dunnet T3 post-hoc test. **p<0.01. (**C, D**) In the fibrin bead assay with HUVECs and HPF-c, afatinib inhibited endothelial cell (EC) sprouting induced by co-stimulation with rhVEGF-A and rhTGFA. Representative images of each group are presented at d7 (**C**) ECs are labeled with phalloidin (red) and nuclei with DAPI (blue). Scale bars, 100 μm. Quantitative analysis for the number of sprouts per bead (**D**) was performed with ImageJ software (30 beads/group). Afatinib treatment was started at d3 after HUVECs had already formed angiogenic sprouts. The data from two independent experiments done in triplicates are presented as mean and SEM. One-way ANOVA with Bonferroni post-hoc test. *p<0.05; ***p<0.001.

The online version of this article includes the following source data for figure 6:

**Source data 1.** Raw data for western blot images.

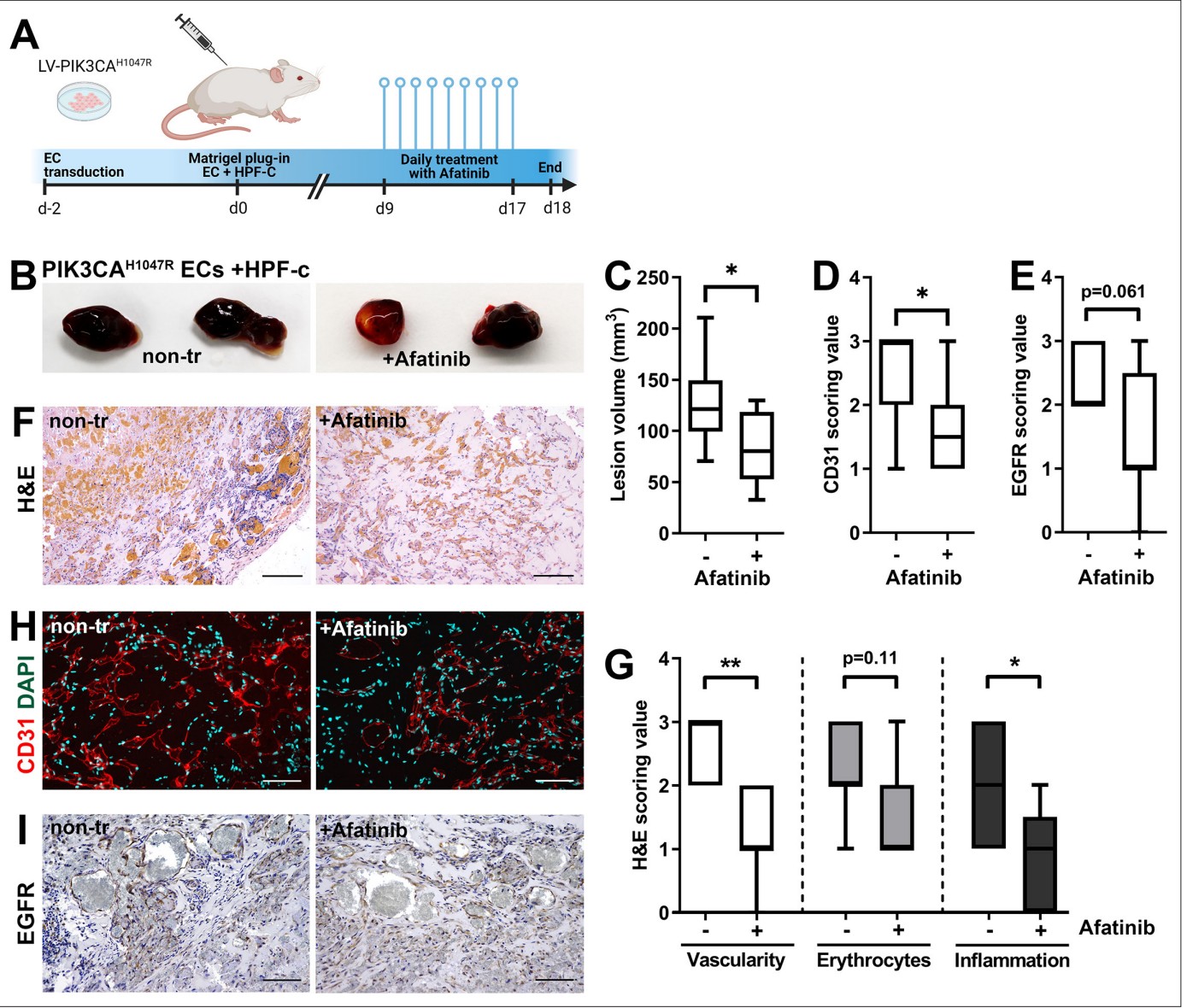

**Figure 7.** Afatinib reduces lesion size in matrigel plug-in assay. Subcutaneous injection of matrigel with HUVECs transfected with *PIK3CA*[H1047R] expressing lentivirus vectors with primary fibroblasts was performed in athymic *Nude-Foxn1[nu]* mice. After lesions reached 200 μm³ in size, afatinib treatment was started for 9 days (25 mg/kg, p.o., daily). (**A**) A timeline of the animal experiment is presented. (**B**) Representative images of dissected explants on day 18. (**C**) Lesion volume measured from dissected explants at d18 (n=7 untreated, n=9 afatinib treated). (**D**) Scoring for vascularization done for sections stained for CD31 (**H**, n=7 untreated, n=9 afatinib treated). (**E**) Scoring for epidermal growth factor receptor (EGFR) expression (**I**); n=6 untreated, n=9 afatinib treated. (**F,H**) Explant sections stained with hematoxylin and eosin (H&E; **F**) or endothelial cell (EC) marker CD31 (red) and DAPI (nuclei, blue; **H**). Scale bars, 200 μm, H&E; 100 μm, CD31. (**G**) Scoring for vascularization, erythrocytes, and inflammatory cells done on H&E-stained sections. (**I**) Explant sections stained with EGFR. Scale bars, 100 μm. Statistical analysis was performed using (**C**) two-tailed unpaired t-test with Welch's correction or (**D, E**) two-tailed Mann-Whitney U test. *p<0.05, and ""p<0.01.

growth, our modified mouse xenograft model was used with ECs expressing oncogenic *PIK3CA*[H1047R] and genotypically normal primary fibroblasts. Lesions were allowed to form for 9 days, reaching 200.1 ± 10.2 μm³ in size, followed by afatinib treatment daily p.o. for 9 days (*Figure 7A*). At d18, lesion size (*Figure 7B–C*) and vascularization detected by H&E and CD31 stainings (*Figure 7H–L*) was shown to be reduced in afatinib-treated mice in comparison to untreated mice. In both groups, various-sized vascular channels filled with erythrocytes were seen (*Figure 7J–K*). A lower number of inflammatory cells was detected in the afatinib treatment group (*Figure 7K*). Accordingly, a reduction of EGFR expression was detected in explants of afatinib-treated mice (*Figure 7 M*). Altogether, the data

validates that ErbB signaling plays a key role in *PIK3CA*[H1047R] lesion formation in the presence of fibroblasts and provides a new potential therapeutic strategy for targeting vascular lesions with a fibrous component.

## Discussion

Symptomatic AST and VM are primarily treated with compression garments, and if needed, with percutaneous sclerotherapy, percutaneous cryotherapy, endovascular laser treatment, or surgical resection (*Rosenblatt, 2007*; *Steiner et al., 2013*; *Hage et al., 2018*; *Cornelis et al., 2017*; *Patel et al., 2017*). In sclerotherapy, sclerosants are administrated using ultrasound guidance intravenously to induce endothelial damage that leads to a total or partial atrophy of the lesion. Single sclerotherapy rarely results in an adequate treatment response but often reduces the lesion size alleviating the symptoms (*Veräjänkorva et al., 2016*). In our earlier study, insufficient response to sclerotherapy was detected primarily in patients with lower-extremity intramuscular AST lesions (*Aronniemi et al., 2017*). Thus, surgery has been used as the primary treatment for AST, whereas percutaneous sclerotherapy is often the first treatment option for VM. Due to high recurrence, difficult anatomical location, the possible functional impairment associated with the operation, and a risk of tissue necrosis after sclerotherapy, more effective therapies are, however, needed for the treatment of AST and VM. Previously, mTOR inhibitor sirolimus has been tested in clinical trials with promising results for patients having VM and somatic mutations leading to constitutive activation of the PI3K pathway [19,20] (ClinicalTrials.gov, study nro: NCT02638389).

In sporadic VM, PI3K/Akt activating somatic *TEK* mutations are associated with skin lesions, and p.L914F mutation is found in 60% of the patients (*Limaye et al., 2015*; *Soblet et al., 2013*). Genetics and disease mechanisms in non-skin-associated VMs are less defined. VMs with somatic mutations in *PIK3CA* do not extend to the skin (*Limaye et al., 2015*) and are found in approximately 20% of the patients. In the study of *Castel et al., 2016*, 24% of the patients (4/17) having intramuscular sporadic VM lesions had a mutation in the *PIK3CA* gene, and 2 out of 17 patients in *TEK* p.L914F (*Castel et al., 2016*). Prior to our study, in AST, only seven patients have been confirmed to have oncogenic *PIK3CA* variants, of these three patients had *PIK3CA* p.H1047R variant, 3 *PIK3CA* p.E542K variant, and one *PIK3CA* p.E545K variant (*Boccara et al., 2020*). Our study is the first to demonstrate *PIK3CA* mutations also in ECs isolated from AST. Oncogenic *PIK3CA* variants were detected in our study in the majority of AST lesions (75%, 15/20 patients), supporting thus the finding of *Boccara et al., 2020* and the importance of oncogenic *PIK3CA* mutation in AST lesion formation. Additionally, we detected in this study a novel somatic mutation in *PIK3CA*, p.H1047L, in AST.

Besides genotypically abnormal ECs, other cell types of venous lesions could affect angiogenic phenotype of ECs e.g., via secretion of paracrine growth factors and thus, contributing to lesion formation. We demonstrate here for the first time that TGFA, a known pro-angiogenic growth factor (*De Luca et al., 2008*; *Leker et al., 2009*; *Schreiber et al., 1986*), is upregulated in VM and AST lesions, and in the presence of an oncogenic *PIK3CA* variant. TGFA and its receptor EGFR are located in both intervascular stromal cells and the endothelium. We further demonstrated that patient SCs were able to: (i) secrete TGFA and VEGF-A; and (ii) transform genotypically normal ECs toward a pro-angiogenic phenotype. Please see *Figure 8* for the proposed model for the paracrine signaling of TGFA/VEGF-A in vascular lesions. Accordingly, our experiments in a modified mouse xenograft model showed an increase in lesion vascularization and erythrocyte amount when genotypically normal fibroblasts were used together with human ECs expressing *PIK3CA* isoforms. We also demonstrated that afatinib, an irreversible inhibitor of EGFR/ErbB1, ErbB2, and ErbB4, was able to decrease *PIK3CA*[H1047R]-induced lesion growth and vascularization. As ErbB signaling has been shown to induce activation of RAS/MAPK and PI3K/Akt pathways that are involved in cell proliferation and inhibition of apoptosis (*Zong et al., 2017*), targeting of both ECs and intervascular SCs by pharmacological agents could be beneficial to increase treatment response in patients with VM or AST having a fibrous component.

Previously, hypoxic avascular stromal cells were suggested to regulate angiogenesis. For example, cancer-associated fibroblasts can induce tumor initiation, progression, and angiogenesis by producing growth factors, proteases, chemokines, and extracellular matrix (*Kalluri, 2016*; *Watnick, 2012*). Fibroblasts have also been shown to modulate EC/pericyte migration, and to be crucial for lumen formation. They are also the main source of VEGF-A production in cancer (*Hughes, 2008*). Perturbation of VEGFR signaling is linked to most vascular anomalies and has been demonstrated for example in

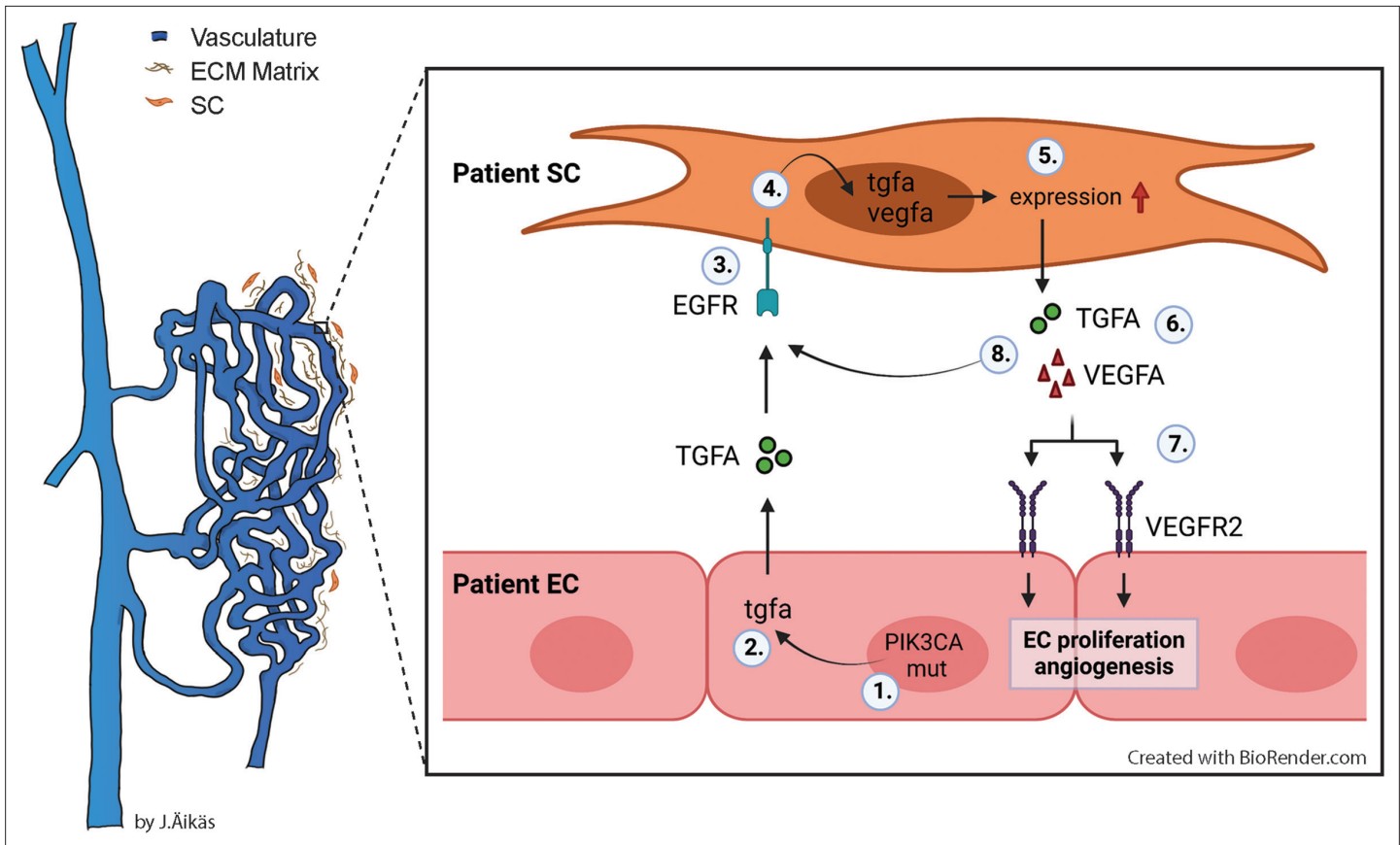

**Figure 8.** Proposed model for the paracrine signaling of TGFA/VEGF-A in the vascular lesion. Schematic illustration showing the general structure of venous malformation or angiomatosis of soft tissue. Pathological vasculature in the lesion (dark blue) is surrounded by a disorganized extracellular matrix (ECM) and intervascular stromal cells (SCs, orange). High magnification from the area close to the vessel wall demonstrates the proposed model for crosstalk between endothelial cells (ECs) and SCs. A mutation in phosphatidylinositol-4,5-biphosphate 3-kinase catalytic subunit alpha (*PIK3CA*) gene (1) or other processes promote ECs to express the high level of transforming growth factor A (TGFA) (2). TGFA binds to epithelial growth factor receptor (EGFR) on the surface of adjacent SCs (3). Activated EGFR-downstream signaling (4) promotes elevated expression of vascular endothelial growth factor (VEGF)-A in SCs and increases the expression of TGFA (5). VEGF-A secreted from SCs (6) binds to VEGF-receptor-2 (VEGFR2) on the surface of ECs (7) and together with TGFA activates angiogenic EC phenotype. TGFA secreted from the SCs (8), can further activate EGFR and its downstream signaling.

infantile hemangioma and arteriovenous malformation (*Chang et al., 1999*; *Koizumi et al., 2002*). To our knowledge, no study has reported the role of VEGF-A or TGFA in VM or AST. A comparison of various vascular anomalies is needed to understand the possible diagnostic significance of TGFA/EGFR expression in VM and AST.

Current treatment strategies targeting cancer-associated fibroblasts aim to: (i) inhibit secretion of pro-angiogenic growth factors; (ii) reduce the accumulation of cells to tumor microenvironment via anti-fibrotic agents; and (iii) inhibit expression of lysyl oxidase-like proteins that regulate ECM integrity (*Puré and Lo, 2016*). Whereas cancer cells are considered genetically instable by accruing mutations that allow escape from cellular regulatory mechanisms and enable the development of drug resistance, cancer-associated stromal cells are not typically mutated in cancer. In VM, mutations in *PIK3CA* or *TEK* genes have been shown to occur solely in EC fraction (*Goines et al., 2018*). We also detected *PIK3CA* mutations only in the EC fraction of AST lesions. Besides the clear role of these mutations in ECs driving the lesion formation, we here demonstrate that lesion-derived intervascular SCs are able to secrete pro-angiogenic growth factors that can change genotypically normal EC function and enable angiogenesis. Besides genetic factors, a hypoxic environment has been shown to induce overexpression of EGFR in cancer (*Franovic et al., 2007*) and to upregulate both TGFA and VEGF-A in cancer and ECs (*Gunaratnam et al., 2003*; *Lee et al., 2018*). In addition, depletion of both HIF1A and HIF2A has previously been shown to lead to the downregulation of TGFA expression (*Chang et al., 2013*).

In our study, *HIF*s and their transcriptional target, *VEGFA*, were found to have a strong or moderate positive correlation with *TGFA* mRNA in patient lesions. *TGFA* expression was also shown to be upregulated in the presence of an oncogenic *PIK3CA* variant, and common transcriptional targets for patient ECs and *HIF*s or *PIK3CA*-expressing ECs was detected. As some of the patients used in this study had received sclerotherapy, the treatment may have caused a hypoxic environment of cells. To conclude, we have identified, for the first time, the involvement of TGFA in vascular lesions and demonstrated the role of fibroblasts in mediating lesion growth and angiogenesis. Targeting of intervascular SCs together with ECs could be beneficial for the treatment of VM and AST with fibrous connective tissue and needs further assessment.

## Acknowledgements

This study was supported by grants from the Academy of Finland (328835, 321535 and 353376 JPL; 287478 and 294073 MUK), Ella and Georg Ehnrooth foundation (JPL), CoE of Cardiovascular and Metabolic Disease (307402, SYH), GeneCellNano Flagship Program (337120 SYH and JPL), the ERC grants (GA670951 SYH and 802825 MUK), Sigrid Jusélius Foundation (MUK, SYH), Finnish Foundation for Cardiovascular Research (MUK, SYH, JPL), Jane and Aatos Erkko Foundation (MUK) and Department of Musculoskeletal and Plastic Surgery, Helsinki University Hospital (PV). Authors thank Gordon Mills & Kenneth Scott for providing pHAGE-PIK3CA and pHAGE-PIK3CA-H1047R plasmids; National Virus Vector Laboratory (the University of Eastern Finland, AI Virtanen Institute, Kuopio, Finland) for producing the lentiviral vectors; Single Cell Genomics Core (the University of Eastern Finland, AI Virtanen Institute, Kuopio, Finland) for preparing and sequencing RNAseq libraries, UEF Cell and Tissue Imaging Unit (the University of Eastern Finland, Biocenter Kuopio and Biocenter Finland, Kuopio, Finland) for the support on Confocal imaging and experiments with Incucyte; and the personnel at the Kuopio University Hospital maternity ward (Kuopio, Finland) for providing umbilical cords for HUVEC extraction.

## Additional information

### Funding

| Funder | Grant reference number | Author |
| --- | --- | --- |
| Academy of Finland | 328835 | Johanna P Laakkonen |
| Academy of Finland | 321535 | Johanna P Laakkonen |
| Academy of Finland | 353376 | Johanna P Laakkonen |
| Academy of Finland | 287478 | Minna U Kaikkonen |
| Academy of Finland | 294073 | Minna U Kaikkonen |
| Ella and Georg Ehnrooth Foundation | | Johanna P Laakkonen |
| CoE of Cardiovascular and Metabolic Disease | 307402 | Seppo Ylä-Herttuala |
| GeneCellNano Flagship Program | 337120 | Seppo Ylä-Herttuala Johanna P Laakkonen |
| European Research Council | GA670951 | Seppo Ylä-Herttuala |
| European Research Council | 802825 | Minna U Kaikkonen |
| Sigrid Jusélius Foundation | | Minna U Kaikkonen Seppo Ylä-Herttuala |
| Finnish Foundation for Cardiovascular Research | | Minna U Kaikkonen Seppo Ylä-Herttuala Johanna P Laakkonen |

| Funder | Grant reference number | Author |
|---|---|---|
| Jane and Aatos Erkko Foundation | | Minna U Kaikkonen |
| Department of Musculosceletal and Plastic Surgery, Helsinki University Hospital | | Pia Vuola |

The funders had no role in study design, data collection and interpretation, or the decision to submit the work for publication.

## Author contributions

Suvi Jauhiainen, Conceptualization, Data curation, Formal analysis, Validation, Investigation, Visualization, Methodology, Writing - original draft, Writing – review and editing; Henna Ilmonen, Data curation, Formal analysis, Validation, Investigation, Visualization, Methodology, Writing - original draft, Writing – review and editing; Pia Vuola, Conceptualization, Resources, Data curation, Formal analysis, Investigation, Writing - original draft, Project administration, Writing – review and editing; Heta Rasinkangas, Data curation, Formal analysis, Validation, Investigation; Heidi H Pulkkinen, Data curation, Formal analysis, Validation, Investigation, Methodology; Sara Keränen, Data curation, Formal analysis, Investigation, Visualization, Writing - original draft; Miika Kiema, Formal analysis, Validation, Investigation, Methodology; Jade J Liikkanen, Formal analysis, Investigation, Visualization, Methodology; Nihay Laham-Karam, Formal analysis, Investigation, Methodology, Writing – review and editing; Svetlana Laidinen, Validation, Investigation, Visualization, Methodology; Mustafa Beter, Software, Formal analysis, Performed quantitative analysis by SproutAngio tool; Einari Aavik, Resources, Data curation, Formal analysis, Writing – review and editing; Kimmo Lappalainen, Jouko Lohi, Resources, Data curation, Formal analysis, Investigation; Johanna Aronniemi, Resources, Data curation, Formal analysis, Investigation, Writing – review and editing; Tiit Örd, Software, Formal analysis, Investigation, Visualization, Methodology, Writing - original draft, Writing – review and editing; Minna U Kaikkonen, Resources, Formal analysis, Investigation, Visualization, Methodology, Writing – review and editing; Päivi Salminen, Resources, Data curation, Formal analysis, Investigation, Project administration, Writing – review and editing; Erkki Tukiainen, Resources, Data curation, Formal analysis, Investigation, Project administration; Seppo Ylä-Herttuala, Resources, Supervision, Project administration, Writing – review and editing; Johanna P Laakkonen, Conceptualization, Resources, Data curation, Formal analysis, Supervision, Funding acquisition, Validation, Investigation, Visualization, Methodology, Writing - original draft, Project administration, Writing – review and editing

## Author ORCIDs

Suvi Jauhiainen http://orcid.org/0000-0002-0630-8175
Miika Kiema http://orcid.org/0000-0001-6438-753X
Nihay Laham-Karam http://orcid.org/0000-0001-8718-4612
Einari Aavik http://orcid.org/0000-0003-3018-9521
Minna U Kaikkonen http://orcid.org/0000-0001-6294-0979
Seppo Ylä-Herttuala http://orcid.org/0000-0001-7593-2708
Johanna P Laakkonen http://orcid.org/0000-0002-8556-9727

## Ethics

Human subjects: Patient sample collection was approved by the Ethical Committee of the Helsinki University hospital, Helsinki, Finland (Decision No 127/13/03/02/2010 and No 1394/2020). Control sample collection was approved by the Research Ethics Committee of the Northern Savo Hospital District, Kuopio, Finland (Decision No 139/2015). Umbilical cord collection for HUVEC isolation was performed with approval from the Research Ethics Committee of the Northern Savo Hospital District, Kuopio, Finland (Decision No 341/2015). Informed consent, and consent to publish, was obtained from all patients included in the study.

Animal experiments were approved by National Experimental Animal Board of Finland (Decision No Esavi-2019-004672) and carried out in accordance with guidelines of the Finnish Act on Animal Experimentation.

**Decision letter and Author response**
Decision letter https://doi.org/10.7554/eLife.82543.sa1
Author response https://doi.org/10.7554/eLife.82543.sa2

## Additional files

### Supplementary files
- Supplementary file 1. NGS experiments.
- Supplementary file 2. gBlock gene fragments.
- MDAR checklist

### Data availability
RNA-seq data has been submitted to NCBI Gene Expression Omnibus under accession numbers GSE130807 and GSE196311.

The following datasets were generated:

| Author(s) | Year | Dataset title | Dataset URL | Database and Identifier |
|---|---|---|---|---|
| Laakkonen JP, Kaikkonen MU, Örd T | 2022 | Gene expression profiling of HUVEC-s expressing PIK3CA with H1047R point mutation | https://www.ncbi.nlm.nih.gov/geo/query/acc.cgi?acc=GSE196311 | NCBI Gene Expression Omnibus, GSE196311 |
| Laakkonen JP, Kaikkonen MU | 2023 | Activation of Epidermal Growth Factor Receptor Pathway in Slow-Flow Vascular Malformations | https://www.ncbi.nlm.nih.gov/geo/query/acc.cgi?acc=GSE130807 | NCBI Gene Expression Omnibus, GSE130807 |

The following previously published dataset was used:

| Author(s) | Year | Dataset title | Dataset URL | Database and Identifier |
|---|---|---|---|---|
| Kaikkonen MU, Downes N | 2018 | RNA expression profiles from HUVECs overexpressing adenovirally delivered HIF1a and HIF2a proteins | https://www.ncbi.nlm.nih.gov/geo/query/acc.cgi?acc=GSE98060 | NCBI Gene Expression Omnibus, GSE98060 |

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

# Appendix 1

## Appendix 1—key resources table

| Reagent type (species) or resource | Designation | Source or reference | Identifiers | Additional information |
|---|---|---|---|---|
| Strain, strain background (*Mus musculus*, female) | Hsd: Athymic Nude-Foxn1$^{nu}$ | Envigo | | Autosomal recessive mutation on nu locus on chromosome 11; T-cell deficient; accepts xenograft transplantation. |
| Cell line (*Homo sapiens*) | HUVEC | This paper | | Umbilical cord endothelial cells isolated in house; see materials and methods section |
| Cell line (*Homo sapiens*) | HSaVEC | PromoCell | # C-12231 | Saphenous vein endothelial cells |
| Cell line (*Homo sapiens*) | HPF-c | PromoCell | # C-12360 | Pulmonary fibroblast |
| Transfected construct (*Homo sapiens*) | LV-PIK3CA | This paper | PIK3CA$^{wt}$ | Lentiviral construct to transfect and express PIK3CA wild-type; produced using Addgene plasmid #116771 |
| Transfected construct (*Homo sapiens*) | LV-PIK3CA-H1047R | This paper | PIK3CA$^{H1047R}$ | Lentiviral construct to transfect and express PIK3CA with oncogenic point mutation on p.H1047R; produced using Addgene plasmid #116500 |
| Biological sample (*Homo sapiens*) | Patients' AST/VM tissue | This paper | Patients' lesion | Tissue material collected during surgery; see materials and methods section |
| Biological sample (*Homo sapiens*) | Endothelial cells from patients' AST/VM | This paper | Patient EC (CD31+) | Freshly isolated from AST/VM tissue; see materials and methods section |
| Biological sample (*Homo sapiens*) | Stromal cells from patients' AST/VM | This paper | Patient SC (CD31-/Vimentin+) | Freshly isolated from AST/VM tissue; see materials and methods section |
| Recombinant DNA reagent | pHAGE-PIK3CA | Addgene | # 116771 | Plasmid encoding PIK3CA wild-type |
| Recombinant DNA reagent | pHAGE-PIK3CA-H1047R | Addgene | # 116500 | Plasmid encoding PIK3CA with oncogenic point mutation on p.H1047R |
| Sequence-based reagent | siRNA: nontargeting control #1 | Thermo Fisher Scientific | # 4390844 | Silencer Select siRNA |
| Sequence-based reagent | siRNA: nontargeting control #2 | Thermo Fisher Scientific | # 4390847 | Silencer Select siRNA |
| Sequence-based reagent | siRNA: targeting to human TGFA | Thermo Fisher Scientific | Assay ID: s14053 | Silencer select siRNA |
| Sequence-based reagent | PrimePCR ddPCR mutation detection assays for PIK3CA c.3140A>G (p.H1047R) | Bio-Rad | Assay IDs: dHsaMDM5225715851 (mut) and dHsaMDW5225715853 (wt) | |
| Sequence-based reagent | PrimePCR ddPCR mutation detection assays for PIK3CA c.3140A>T (p.H1047L) | Bio-Rad | Assay IDs: dHsaMDM2916088171 (mut) and dHsaMDW2916088173 (wt) | |
| sequence-based reagent | PrimePCR ddPCR mutation detection assays for PIK3CA c.1633G>A (p.E545K) | Bio-Rad | Assay IDs: dHsaMDM9869636521 (mut) and dHsaMDW9869636523 (wt) | |
| sequence-based reagent | PrimePCR ddPCR mutation detection assays for PIK3CA c.1624G>A (p.E542K) | Bio-Rad | Assay IDs: dHsaMDM3010833491 (mut) and dHsaMDW3010833493 (wt) | |
| sequence-based reagent | Custom-design Taqman SNP Genotyping assays for TEK c.2740C>T (p.L914F) | Thermo Fisher Scientific | # 4331349 | fwd 5'-CTTCCCTCCAGGCTACTT-3', rev 5'-AATGCTGGGTCCGTCT-3', reporter 1 (HEX) 5'-CTTGCGAA GGAAGTCCAGAAGGTTTC-3', and reporter 2 (FAM) 5'- CTTG CGAAAGAAGTCCAGAAGGTTTC-3' |
| peptide, recombinant protein | rhTGFA | Sigma-Aldrich | # GF313 | |

*Appendix 1 Continued on next page*

*Appendix 1 Continued*

| Reagent type (species) or resource | Designation | Source or reference | Identifiers | Additional information |
|---|---|---|---|---|
| commercial assay or kit | Human CD31 microbead kit | Miltenyi Biotec | # 130-091-935 | |
| commercial assay or kit | RNeasy Mini Kit | Qiagen | # 74106 | |
| commercial assay or kit | Human Quantikine TGFA ELISA | R&D Systems | # DTGA00 | |
| commercial assay or kit | Human Quantikine VEGF ELISA | R&D Systems | # DVE00 | |
| chemical compound, drug | Afatinib | MedChem Express | # HY-10261 | |
| software, algorithm | SproutAngio | https://github.com/mbbio/SproutAngio (*Beter et al., 2023*) | | Open access tool for quantitation |

