## [Editor Report]

The authors have explored potential crosstalk between endothelial cells and fibroblasts in the context of sporadic vascular malformation (venous malformation and angiomatoses of soft tissue). With a high level of evidence, they found that mutated endothelial cells secrete TGFa that will activate surrounding fibroblasts, leading in turn to VEGFA secretion that will stimulate endothelial cells sprouting and vascular malformation development. This is an important advance in the field.

---

## [Decision Letter]

**Decision letter after peer review:**

Thank you for submitting your article "ErbB Signalling is a Potential Therapeutic Target for Vascular Lesions with Fibrous Component" for consideration by *eLife*. Your article has been reviewed by 2 peer reviewers, and the evaluation has been overseen by a Reviewing Editor and Paul Noble as the Senior Editor. The following individual involved in the review of your submission has agreed to reveal their identity: Guillaume Canaud (Reviewer #2).

Essential revisions:

Please examine the recommendations from Reviewer 1 and respond to the best of your abilities. The reviews are very thoughtful and reasonable

*Reviewer #1 (Recommendations for the authors):*

1. The p.H1047R mutant allelic frequency of the lentivirally-transduced HUVECs must be given.

2. In Figure 4, figure supplement 2, please explain what "fold change in confluency" means – that is how is confluency quantified?

3. The afatinib experiment in Figure 7 should be repeated on xenografts with only ECs because. This would test whether blocking EGFR signaling on the EC is sufficient to block the vascularization. Based on data in Figure 5H, the fibroblasts do not increase the number of perfused vessels formed from the mutant EC. This could perhaps be an indication that the stromal cells are bystanders and not direct contributors to the abnormal vessels.

4. Although perhaps this will be part of a subsequent study, it would be of strong interest to study the VM or AST-derived stromal cells in the xenograft experiments – this would address whether the stromal cells have been altered by being in association with the mutant vessels

*Reviewer #2 (Recommendations for the authors):*

This is a well-written and convincing article. The authors provided a large body of evidence supporting TGFA as an interesting paracrine factor involved in vascular malformation development.

I would recommend adding controls for Figure 2 (healthy skin biopsies).

---

## [Author Response]

Essential revisions:Please examine the recommendations from Reviewer 1 and respond to the best of your abilities. The reviews are very thoughtful and reasonable

We thank the Editors and Reviewers for these valuable comments and have now improved the manuscript accordingly. Please see below our point-by-point response.

Reviewer #1 (Recommendations for the authors):1. The p.H1047R mutant allelic frequency of the lentivirally-transduced HUVECs must be given.

As described above, we have now analyzed fractional abundance of the PIK3CA p.H1047R in matrigel plugin tissue. The data indicates that our lentiviral construct for PIK3CA H1047R is functional and mutated cells are present in the xenograft tissue until the end-point of the experiment.

2. In Figure 4, figure supplement 2, please explain what "fold change in confluency" means – that is how is confluency quantified?

Cellular growth was monitored using IncuCyte Live-Cell Imaging system. Images were acquired in 3-h intervals, 4 images/well, for a 48-h period using a 10x objective. Layer mask to cover cell-containing areas of each image was created using Incucyte software. Based on this the mean confluency of the cells at each time point was calculated automatically. To calculate the fold change in cell confluency, mean confluency of each time point was divided by the mean confluency detected at 0h timepoint (i.e. after mixing the cells together).

Detailed description about quantifications is now added in the manuscript, in section p. 11, rows 214-217.

3. The afatinib experiment in Figure 7 should be repeated on xenografts with only ECs because. This would test whether blocking EGFR signaling on the EC is sufficient to block the vascularization. Based on data in Figure 5H, the fibroblasts do not increase the number of perfused vessels formed from the mutant EC. This could perhaps be an indication that the stromal cells are bystanders and not direct contributors to the abnormal vessels.

We thank the Reviewer for this suggestion. Afatinib treatment was performed only with PIK3CA H1047R EC + FB group as our data strongly supports the importance of stromal cells/fibroblasts as a player promoting formation of pathological vasculature. Please see description above (Public review, comment #3). The role of fibroblasts in tumor angiogenesis have also been shown previously in various studies, and they are known to induce EC proliferation.

Afatinib is an inhibitor of ErbB1/EGFR, ErbB2 and ErbB4. Thus it is not specific for EGFR. As all of these ErbB ligands are known to regulate cell proliferation and angiogenesis, either directly or in a secondary manner, and TGFA was not the only ErbB ligand shown to be upregulated in our RNAseq data from patient-derived CD31+ ECs, afatinib would be highly interesting treatment for inhibiting PIK3CA-driven vascular lesion growth in patients. Further studies on patient samples and patient derived cells are warranted. Although we believe afatinib affects also on vasculature in the matrigel plugs containing PIK3CAmut ECs only, at least EGFR, the receptor of TGFA, was more strongly expressed in fibroblasts in our PIK3CA-driven xenograft lesions, and also prominent in intervascular areas in AST patient lesions. We also strongly believe that PIK3CA H1047R EC + FB combination group is closer to the lesion microenvironment in patients, where fibrotic component is often present, especially on VM patients who have underwent routine treatment of vascular lesion with sclerosing agent without good response or AST, known to have fibroblasts present in the lesion. As written in our manuscript (p. 21, rows 483-488), development of additional treatments would be beneficial especially for the patients not responsive for the routine treatment with sclerotherapy or current therapies aiming to target ECs only.

4. Although perhaps this will be part of a subsequent study, it would be of strong interest to study the VM or AST-derived stromal cells in the xenograft experiments – this would address whether the stromal cells have been altered by being in association with the mutant vessels

We thank the Reviewer for this excellent suggestion and agree that this would be a very interesting experiment. Unfortunately, however, xenograft mouse model requires a very large amount of cultured cells that is not at the moment doable from our patient sample isolations without dividing cells for weeks thus leading to very high cell passages. As primary cells loose quickly their characteristics on a cell culture plate we opted to use commercially available cells.

Reviewer #2 (Recommendations for the authors):This is a well-written and convincing article. The authors provided a large body of evidence supporting TGFA as an interesting paracrine factor involved in vascular malformation development.

I would recommend adding controls for Figure 2 (healthy skin biopsies).

Tissue location of the VM and AST lesions included in the study were:

Intramuscular, 42.1 % of lesions (n=16)Intramuscular and subcutaneous, 21.1 % of lesions (n=8)Intramuscular, subcutaneous and synovial membrane, 5.3 % of lesions (n=2)Intramuscular and synovial membrane, 2.6 % of lesions (n=1)Subcutaneous and synovial membrane, 2.6 % of lesions (n=1)Subcutaneous only, 26.3 % of lesions (n=10)Skin, none of the lesions

Thus we considered normal skeletal muscle to be more representative control tissue for this patient cohort than healthy skin biopsies. Please see presentative images of IHC for TGFA on lesion samples vs control skeletal muscle in Figure 2 —figure supplement 2B.